# High-sensitive spatially resolved T cell receptor sequencing with SPTCR-seq

Jasim Kada Benotmane[1,2,3], Jan Kueckelhaus[1,2,3], Paulina Will[1,2,3], Junyi Zhang [1,2,3], Vidhya M. Ravi [1,2,3,4], Kevin Joseph [1,2,3,4,5], Roman Sankowski [6], Jürgen Beck[1,2], Catalina Lee-Chang [7], Oliver Schnell[1,2,4] & Dieter Henrik Heiland [1,2,3,7,8] ✉

Spatial resolution of the T cell repertoire is essential for deciphering cancer-associated immune dysfunction. Current spatially resolved transcriptomic technologies are unable to directly annotate T cell receptors (TCR). We present spatially resolved T cell receptor sequencing (SPTCR-seq), which integrates optimized target enrichment and long-read sequencing for highly sensitive TCR sequencing. The SPTCR computational pipeline achieves yield and coverage per TCR comparable to alternative single-cell TCR technologies. Our comparison of PCR-based and SPTCR-seq methods underscores SPTCR-seq's superior ability to reconstruct the entire TCR architecture, including V, D, J regions and the complementarity-determining region 3 (CDR3). Employing SPTCR-seq, we assess local T cell diversity and clonal expansion across spatially discrete niches. Exploration of the reciprocal interaction of the tumor microenvironmental and T cells discloses the critical involvement of NK and B cells in T cell exhaustion. Integrating spatially resolved omics and TCR sequencing provides as a robust tool for exploring T cell dysfunction in cancers and beyond.

Spatial transcriptomics has emerged as a powerful tool for studying the complex interactions between cells within their native tissue context, providing valuable insights into cellular heterogeneity, functional states, and the organization of different cell types in the tissue microenvironment[1–3]. This technology has led to significant advancements in understanding diverse biological processes, including development, tissue homeostasis, and disease progression[4–7]. We recently reported on the topography of cellular interactions across tumor, myeloid, and lymphoid cells by integrating spatially resolved transcriptomics, metabolomics, and proteomics[4]. However, a critical unmet need in this field is the spatial mapping of T cell receptors (TCRs), which play a pivotal role in the adaptive immune response

against cancer and other diseases. While effective in providing a global view of gene expression patterns, the current array-based spatially resolved transcriptomic technologies are limited in their ability to annotate TCRs directly. Although the 3′ cDNA sequencing used in array-based spatial transcriptomics is adequate to enumerate mRNA abundance, direct sequencing of recombined TCR genes is hindered since the highly variable CDR3 segment of the TCR is located closer to the 5′ end, not captured by 3′ library preparation[8,9]. The T-cell repertoire is a crucial component of the adaptive immune system, as it recognizes and eliminates infected, damaged, or malignant cells. TCRs are generated through a highly dynamic and stochastic process known as V(D)J recombination, which leads to a virtually limitless diversity of

[1]Department of Neurosurgery, Medical Center - University of Freiburg, Freiburg, Germany. [2]Faculty of Medicine, Freiburg University, Freiburg, Germany. [3]Microenvironment and Immunology Research Laboratory, Medical Center—University of Freiburg, Freiburg, Germany. [4]Translational NeuroOncology Research Group, Medical Center—University of Freiburg, Freiburg, Germany. [5]Center for NeuroModulation (NeuroModul), University of Freiburg, Freiburg, Germany. [6]Institute of Neuropathology, Medical Center—University of Freiburg, Freiburg, Germany. [7]Department of Neurological Surgery, Northwestern University Feinberg School of Medicine, Chicago, IL, USA. [8]German Cancer Consortium (DKTK), partner site Freiburg, Freiburg, Germany. ✉e-mail: dieter.henrik.heiland@uniklinik-freiburg.de

TCR sequences. Each TCR is composed of two chains, α/β or γ/δ respectively, which together form the antigen-binding site responsible for recognizing a specific peptide-MHC complex. The diversity of TCRs arises from the unique combination of variable, diverse, and joining TCR exons, with the majority of this diversity concentrated in the complementarity-determining regions (CDRs). The highest diversity is observed within the epitope binding region of the TCR, CDR3, making it an ideal natural barcode for studying T-cell clonality and diversity[10,11]. Understanding T cells' spatial distribution and functional states and their receptors is essential for unraveling the mechanisms underlying cancer-associated immune dysfunction, discovering hidden features in the TCR sequence and designing effective immunotherapies.

In this study, we present an approach called spatially resolved T cell receptor sequencing (SPTCR-seq), enabling the spatial mapping of TCRs with unprecedented accuracy and resolution. We leverage the SPTCR-seq technology to characterize T-cell clonality and diversity in glioblastoma, the most malignant tumor of the central nervous system. In our comparative analysis of SPTCR-seq with other well-established methodologies, we showcase SPTCR-seqs improved TCR Detection performance and sensitivity due to its enhanced ability to reconstruct the entire TCR, including VDJ rearrangement and the CDR3 region. GBMs are well known for their "cold" immune environment and lack of anti-tumor immunity. Our findings demonstrate that the integration of array-based spatially resolved transcriptomics with SPTCR-seq allows for identifying distinct T cell subpopulations residing in spatially seg-regated niches within the tumor microenvironment. This in-depth characterization of the T cell repertoire at a spatial resolution has important implications for understanding the functional states of T cells, their interactions with other immune and non-immune cells, and the overall immune landscape of the tumor.

## Results

### Workflow of SPTCR-seq

Over the past year, two distinct spatial TCR-seq methodologies were developed, Hudson et al. [12] employed a mixture of TRBV forward primers (further referred to as "Hudson protocol") or RNase H-dependent PCR (Slide-TCR-seq[13]) from the TCRα and TCRβ regions (further referred to as "Liu protocol") to enrich for CDR3-containing TCR chain fragments in Slide-seq libraries, followed by short-read Illumina sequencing[12,14,15]. Another technique to enrich TCR fragments is reported by the RAGE-seq protocol[16], which utilizes target enrichment by hybridization of single cell library followed with long-read sequencing on the Oxford Nanopore Technologies (ONT) platform. This allows for direct annotation of VDJ rearrangement and CDR3 region with high yield and coverage. Inspired by the high fraction of fully annotated TCRs achieved with the RAGE-seq protocol[16], we built the SPTCR-seq protocol with optimized hybridization probes and an analysis pipeline for accurate TCR reconstruction, Fig. 1a. SPTCR-seq is compatible with widely-used high-throughput spatially resolved tran-scriptomic technologies, including ST, Visium 10x Genomics, STEREO-seq[17], seq-SCOPE[18], Slide-seq[19], and scRNA-seq platforms (10x Geno-mics 3' protocols, Seq-Well, and Drop-seq). Each of these platforms involves an intermediate step that generates full-length cDNA tran-scripts. From these transcripts, we designed probes targeting regions of all annotated TCRs known to the ImMunoGeneTics information system[20] resulting in 186 probes, each 100 bp in length. This is followed by amplification and nanopore library preparation. Sequencing can be performed at the desired coverage using Flongle for low or MinION/ PromethION for high coverage, Supplementary Figure 1a. After UMI correction, TCR annotation with IGBLAST[21] demonstrated full VDJ-annotations for TCRβ in roughly 60% (~22.36 × 10^6 UMI counts) and TCRα VJ-annotations in 93.8% (33.9 × 10^6 UMI counts). TCRγδ-chains were detected with lower abundance and no complete VDJ- (TCRδ) or VJ-annotations (TCRγ). A co-expression analysis of TCRα and TCRβ revealed a significant correlation coefficient of 0.693 ($p < 2.2 \times 10^{-16}$),

indicating that the TCRαβ-chain distribution aligns with expectations, Fig. 1b, c. To investigate niche-specific T cell exhaustion, we addressed the Visium array technology's 55 μm spot size limitation by employing three distinct algorithms (CytoSpace[22], Cell2location[23]) for further data processing. This approach allowed us to estimate cell type abundance across spots with high concordance to TRBC2 and TCRα/TCRβ chain expression, thereby effectively characterizing cell-specific expression and inferring T cell distribution (Fig. 1d–f).

### SPTCR-seq computational workflow

A notable limitation of the nanopore platform is its base call accuracy for long-read sequencing, with a mean error rate of ~2–4% in R9 pore sequencing experiments[24,25]. This presents significant challenges when attempting to anchor reference regions during VDJ-reconstruction annotation. To mitigate this issue, we developed a computational toolkit compensating for technical biases implicit in long-read sequencing through consensus-based error correction. Building on our previous platform for spatial data analysis (SPATA2), the add-on *SPATAImmune* R-package integrates spatially resolved multi-omics and the T-cell repertoire. We increase the yield of annotated reads by adopting the computational postprocessing using the following steps, Fig. 2a and Supplementary Fig. 2: First, barcode and UMI annotation were performed using the recently developed scTagger[26] pipeline, which utilizes the True-seq read1 as annotation anchors resulting in 89.4% correctly demultiplexed reads. Since nanopore reads lose their orientation, we split, trimmed, and reorientated the reads (pyChopper[27] and cutadapt[28]), followed by TCR annotation using *IGBLAST*[21]. These VDJ annotations were then used for read-correction to enhance the anchor sequences (CDR1/CDR2) alignment required for CDR3 annotation. The reads were then clustered by their VJ-similarity distance and spatial position and further corrected using the RATTLE[29] algorithm in a supervised manner. This approach reduced nanopore error-rate from 4.74 ± 2.49 to 1.11 ±1.43 percent without changing the size of the transcript (mean length uncorrected 820 ± 278 vs corrected 822 ± 277) Fig. 2b, c. We assessed quality improvement by inspecting the mapping identity score of IgBlast, which indicates the number of mismatches of the aligned segment with the reference. The mapping score for the V region of the TCR segment improved from a mean of 59% for uncorrected Nanopore R9 reads to a mean of 80% mapping identity following correction, Fig. 2d. With Read Errors being the result of basecalling or artifacts in the PCR, they lead to an overly diverse TCR Repertoire with numerous TCRs at low read counts. Driven by the hypothesis that, higher error rates would falsely overdiversify the repertoire we used the number of clones with fewer than 50 reads as a proxy for high error rates and found a significant decrease after cor-rection. Consequently, the mean sequencing depth per annotated T cell clone increased 1.86 times from 445 ± 3568 to 829 ± 7677 sequen-cing reads following correction, Fig. 2d. Recent advancements in Nanopore chemistry have demonstrated lower per-base error rates comparable to other long-read sequencing technologies like PacBio, potentially even surpassing Illumina's per-base accuracy due to fewer PCR cycles or complete lack thereof[30]. The primary source of sequencing errors has been attributed to the PCR process itself[30]. To assess the differences between Nanopore chemistries, we compared a set of three samples sequenced on both R9 and R10 flow cells. Our analysis revealed that sequencing on R10 flow cells improved the V-segment mapping score by ~5% compared to R9 flow cells (R9: 59%; R10: 63.4%), presumably mitigating basecalling errors. The SPTCR pipeline then further boosted performance, achieving a high propor-tion of accurately aligned TCR-segments for both R9 and R10. Our findings, in conjunction with recent literature, suggest that the major source of error originates during PCR amplification and accumulates in protocols with increased PCR cycles[30]. We assessed T cell abundance across samples and histological regions (based on the IVY classifica-tion) using UMI-corrected TCRβ expression, which indicates T cell

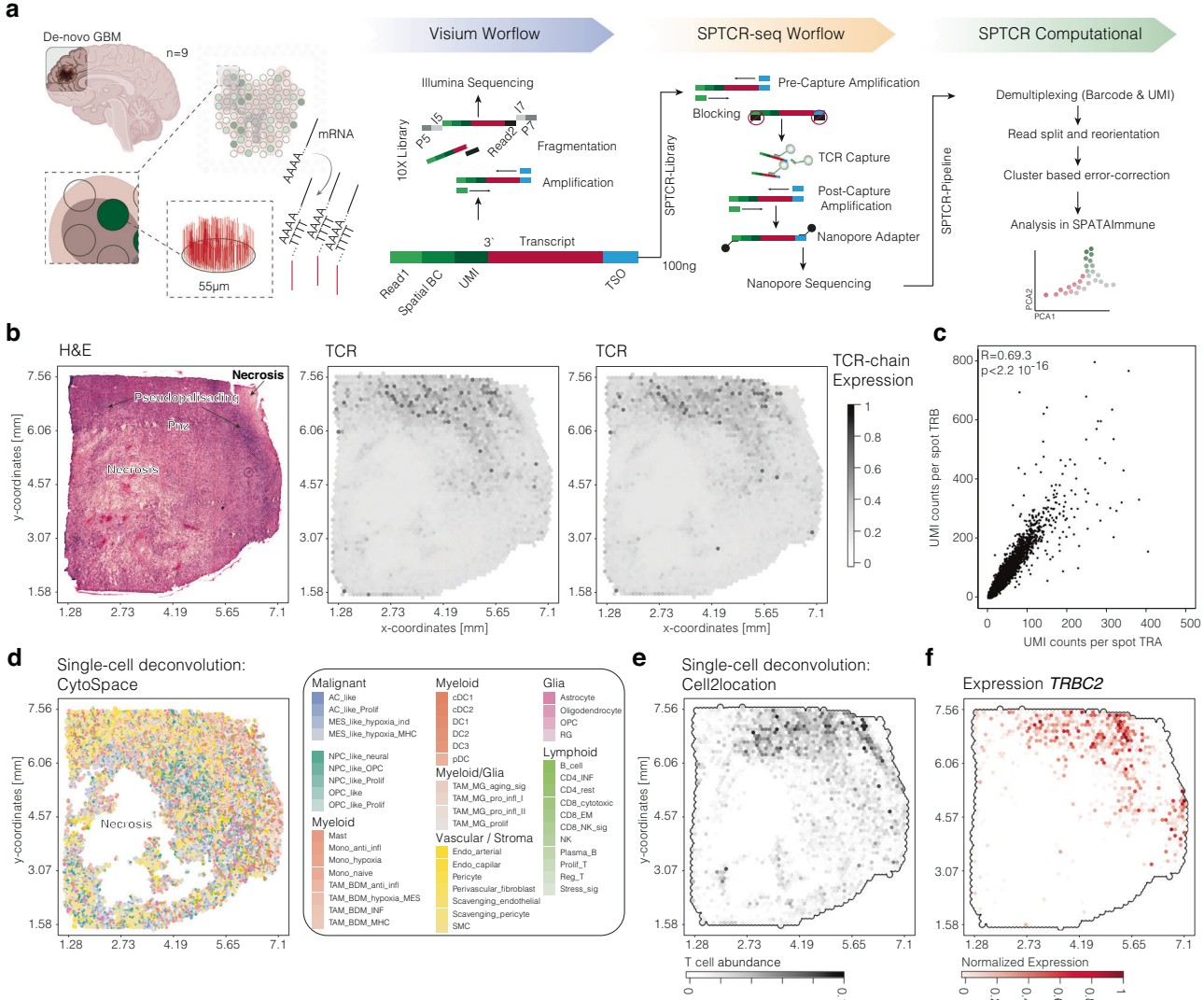

**Fig. 1 | Overview of the SPTCR workflow. a** Illustration of the workflow combining spatially resolved transcriptomics with T cell receptor profiling in the SPTCR-seq protocol. Independent from the standard Visium protocol, the yield of 100 ng full-length cDNA is taken to enrich TCR transcripts through hybridization. After long-read sequencing on the nanopore platform, intensive postprocessing is applied and data will be integrated through the R-based software tool SPATAImmune. **b** H&E histological image of the sample UKF313 with annotation of defined histological regions. Surface plots indicate the abundance of TCR chains. **c** Scatter plot of TRA UMI counts per spot (y-axis) and TRB UMI counts per spot (y-axis). Pearson correlation $R = 0.693$, $p < 2.2 \times 10^{-16}$. **d** Surface plot of the single-cell composition after deconvolution with CytoSPACE22. Surface plot of the T cell abundance using Cell2Location scores (**e**) or the expression of TRBC2 (**f**). Surface plots contain the dimensions of the samples (in mm) on the $x$ and $y$ axes. Partially created with BioRender.com.

presence, as shown in Fig. 2f. In four out of nine samples, TCRβ expression was relatively low, while five samples displayed high T cell abundance with around 30 TCRβ UMIs per spot. We observed a significant enrichment of T cell abundance in perinecrotic, microvascular proliferation and hyperplastic blood vessel regions (ANOVA $p = 0.0043$), with the lowest abundance in infiltrative and leading-edge areas (ANOVA $p = 0.0173$). Using SPTCR-seq, we could obtain a detailed characterization of spatial TCR clonality, which requires sufficient sequencing depth to annotate each clone's spatial expansion. To this end, we quantified the number of UMIs per clone across samples. Our analysis showed that approximately 75% of all clones contained more than one UMI, while 8.8% had over five UMIs, Fig. 2g.

## Comparative analysis of SPTCR-seq and PCR-based Spatial TCR-seq methods
In the next phase of our study, we aimed to compare SPTCR-seq with published protocols to evaluate their respective advantages and disadvantages. Both PCR-based protocols (Hudson and Liu) are

considerably more cost-effective and less labor-intensive than SPTCR-seq: hands-on time for Hudson and Liu protocols is 4–6 h, while for SPTCR-seq, it is 6–8 h plus an overnight incubation. The cost per sample (excluding sequencing) is ~5 Euros for the Hudson protocol, 12 Euros for the Liu protocol (2.4 fold-change), and 32 Euros for SPTCR-seq (6.4 fold-change). It should be noted that the estimated hands-on time and cost per sample may vary if samples are multiplexed together. The primary cost difference can be attributed to the library preparation workflow, sequencing depth and the degree of sample multiplexing, as Nanopore technology tends to be significantly more expensive than Illumina when samples are not multiplexed. To compare the protocols, we used three Visium libraries and performed all three protocols as recently described. The sequencing depth from Hudson and Liu was ~300 million reads/sample, and the SPTCR-seq yield was around 26–30 million reads. All protocols were processed by the individual pipelines, as described in Fig. 3a. We began our comparison by examining the diversity of distinct TCRs detected by each method. To this end, we evaluated the number of unique CDR3s per

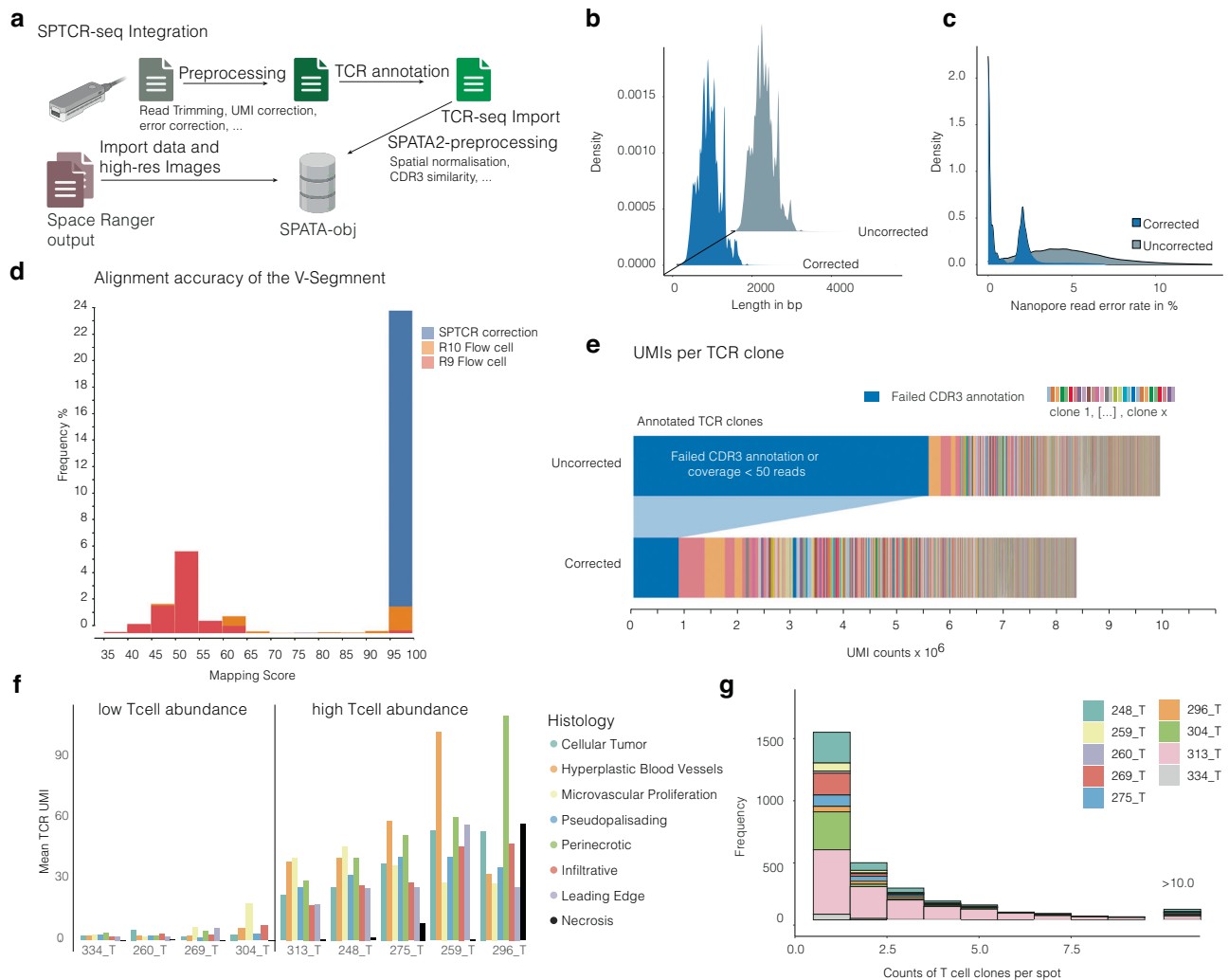

**Fig. 2 | Quality control and validation of VDJ-rearrangement. a** Illustration of the workflow for the SPTCR postprocessing. **b** Density plot of error rates from corrected and uncorrected reads. **c** Density plot of mean transcript length obtained by long-read sequencing before and after error correction. **d** Barplot of the mapping score between R9 (raw), R10 (raw), or SPTCR processed reads. **e** Stacked bar plot of reads annotated to defined T cell clones. Through error correction, the number of reads that either belong to clones with less than 50 reads or in which the CDR3 region was not detectable was significantly reduced ($p = 3.54 \times 10^{-6}$, Chi-square test). **f** A Barplot indicates the number of the mean fully annotated and UMI-corrected TCRs across samples and histological regions. **g** Barplot illustration of the number of T cell clones per spot (x-axis) and its frequency (y-axis). Colors indicate the sample origin. Surface plots contain the dimensions of the samples (in mm) on the x and y axes. Partially created with BioRender.com.

sample by applying a stringent criterion that classified CDR3 regions as unique if they exhibited a Levenshtein distance of at least two. Due to the lack of TCRα chains in the Hudson protocol, we focused our comparison on the TCRβ chain. The Liu protocol identified a mean of 13.4 unique TRBs (min = 4, max = 31, $n = 5$) per sample, whereas the Hudson protocol yielded a mean TRB count of 26, ranging from 16 to 36 TCRs per sample ($n = 3$). In comparison, SPTCR-seq detected a mean of 208 unique TRBs (sd:135) per sample ($n = 5$). For combined TRA & TRB counts, the Liu protocol achieved a mean of 16.2 (min = 6, max = 33, $n = 5$) in matched samples, compared to SPTCR's 208.6 unique TRA/B CDR3 regions (min = 24, max = 343). Our data showed that SPTCR-seq maintained a more diverse immune receptor repertoire, with a mean of 87.25 UMIs per TRB TCR, as opposed to 1.2 UMIs per TCR across all samples for the Hudson protocol ($n = 3$) and Liu protocol ($n = 7$). To investigate the reasons behind the lower number of detected TCRs in PCR-based methods compared to SPTCR-seq, we visualized the mapped segments reported by MixCR and SPTCR-seq in a waterfall plot, Fig. 3b. The TCR reconstruction algorithm successfully mapped 93% of reads to a TCR locus for the Hudson protocol ($n = 3$)

and 56% for the Liu protocol ($n = 7$), while IgBlast mapped SPTCR-seq reads to a TCR locus in 98.4%. As we dropped uncorrected Reads that did not hold a V and J Region prior to correction, SPTCR lost 40% of the reads during alignment. Subsequently, we visualized the spatial distribution of TCRs and observed a consistent spatial abundance pattern of unique TRB UMIs across all methods, Fig. 3c, d. The main observed limitation of PCR-based methods was their inability to accurately annotate J regions (-92% failure rate for the Hudson protocol and 55% for the Liu protocol), which results in a significant loss of reads necessary for complete TCR reconstruction. In contrast, SPTCR-seq successfully annotates V and J regions 87.56% of the time. Although the final anchoring of CDR regions still remains challenging, SPTCR-seq attains a considerably higher rate of full TCR reconstruction compared to other methods. We primarily attribute this significant difference to the shorter read length of the 300 Cycles v3 Illumina chemistry. Although Illumina's high throughput excels at capturing gene expression at high coverage, the short read length hampers its ability to span the structurally important regions of the TCR. In conclusion, our comparison of SPTCR-seq with the Hudson and Liu protocols

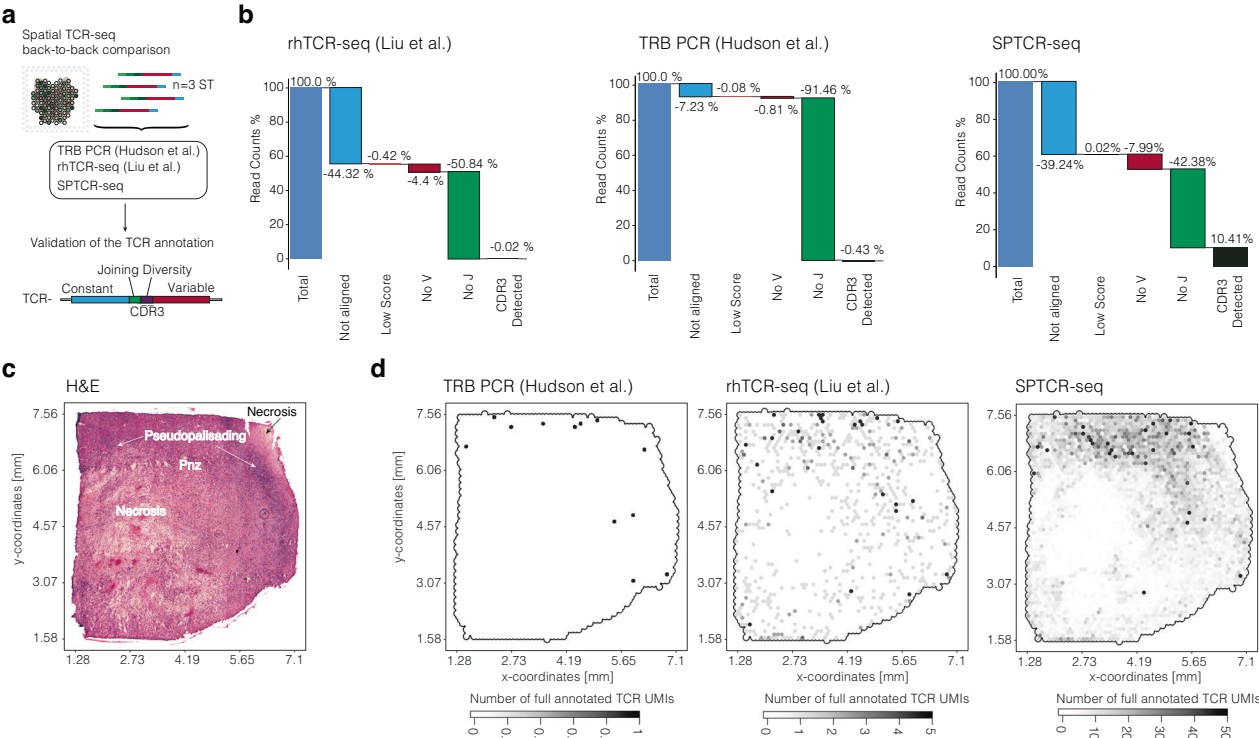

**Fig. 3 | Cross-method comparison of spatial resolved TCR-seq. a** Illustration of the comparison of three methods. **b** Waterfall plots of the three methods indicate the loss of reads during the TCR annotation and reconstruction. Surface plot indicates the histology (**c**) and the spatial distribution of annotated TCRs across methods (**d**). Surface plots contain the dimensions of the samples (in mm) on the *x* and *y* axes.

demonstrated that SPTCR-seq outperforms these PCR-based methods regarding TCR diversity and annotation accuracy. Despite the higher cost and increased hands-on time associated with SPTCR-seq, its sensitivity to detect a more diverse immune receptor repertoire and annotate full TCR transcripts makes it a valuable tool for TCR-seq analysis. Additionally, the robustness and scalability of the SPTCR-seq method make it a promising approach for future studies aiming to examine the spatial distribution and evolution of T cell receptor sequences in various research contexts.

## Identification of spatial niches with clonal T cell expansion or diversity in glioblastoma

Glioblastomas, known as "cold" tumors, exhibit low T cell abundance, significant T cell dysfunction, and tumor-associated exhaustion[7]. Consequently, immunotherapeutic trials frequently fail to achieve desired endpoints[31,32]. Recent single-cell data have highlighted various cellular interactions responsible for the predominantly immunosuppressive tumor microenvironment, with a limited investigation of their spatial occurrence[4,7,33]. Spatially resolved transcriptomics offer the advantage of profiling the transcriptional landscape almost unbiasedly while co-localizing specific cellular functions and interactions. Motivated by the observation that neoadjuvant checkpoint blockade has shown survival differences[34], we explored the possibility of a localized T cell response in the primary tumor within regions typically resected. We employed spatially resolved TCR-seq on nine glioblastoma (GBM) samples to better understand their immune topography, Fig. 4a. Our investigation aimed to identify niches exhibiting local clonal T cell expansion or clonal diversity. We estimated "clonal abundance" using normalized UMI counts per clone and compared it to each clone's spatial distribution, Fig. 4b. We observed significant differences, ranging from strictly local expansion to widely dispersed clones, Fig. 4b. To create an index incorporating clonal abundance within spatial neighborhoods, we constructed a spatial graph using Delaunay

triangulation and estimated each clone's regional enrichment frequency (Clonality-Index). Comparing clonal abundance with the clonality-index allowed us to distinguish clones with local patterns, Fig. 4c. Since many clones shared the same spots, we introduced the "CDR3-diversity-index" parameter, which considers the mean reciprocal Levenshtein distance of clones at a given spot. A high CDR3-diversity-index at the spot level indicates the presence of multiple clones with notably different CDR3 regions, Fig. 4d–f. According to this, we classified spots local T cell repertoire as either expanded or diverse.

## Local clonal diversity is associated with tumor-associated T-cell dysfunction/exhaustion

Utilizing our established parameters, we proceeded to categorize each spot as "local T cell expansion", "local T cell diversity" and "no detected T cells" or spots that could not be unambiguously classified, Fig. 5a. In the analysis of all samples, clonal T cell expansion was observed in only 6 out of 9 samples, in a minority of spots ranging from 2 to 68 spots, while T cell diversity was detected in every sample, Fig. 5b. Furthermore, no distinct histological regions could be associated with local clonal expansion or clonal diversity, Fig. 5c. We postulated that the local expansion of clones is strongly correlated with regions with potential antigens or favorable microenvironmental conditions. Since unique clonal expansion was only rarely observed, indicating that T cells in glioblastoma do not display a single dominating T cell clone; instead, they exhibit T cell expansion independent of a specifically presented antigen. This hypothesis is substantiated by the spatial annotation of clones exhibiting a high Clonality-Index, which revealed a significant overlap of local T cell expansion within the same tumor regions, Fig. 5e, f. Based on these findings, we aimed to investigate to what extent these clones are functionally active to support anti-tumor immunity or dysfunctional/exhausted. Using supervised spatial clustering based on predefined T cell cytotoxic and dysfunctional/

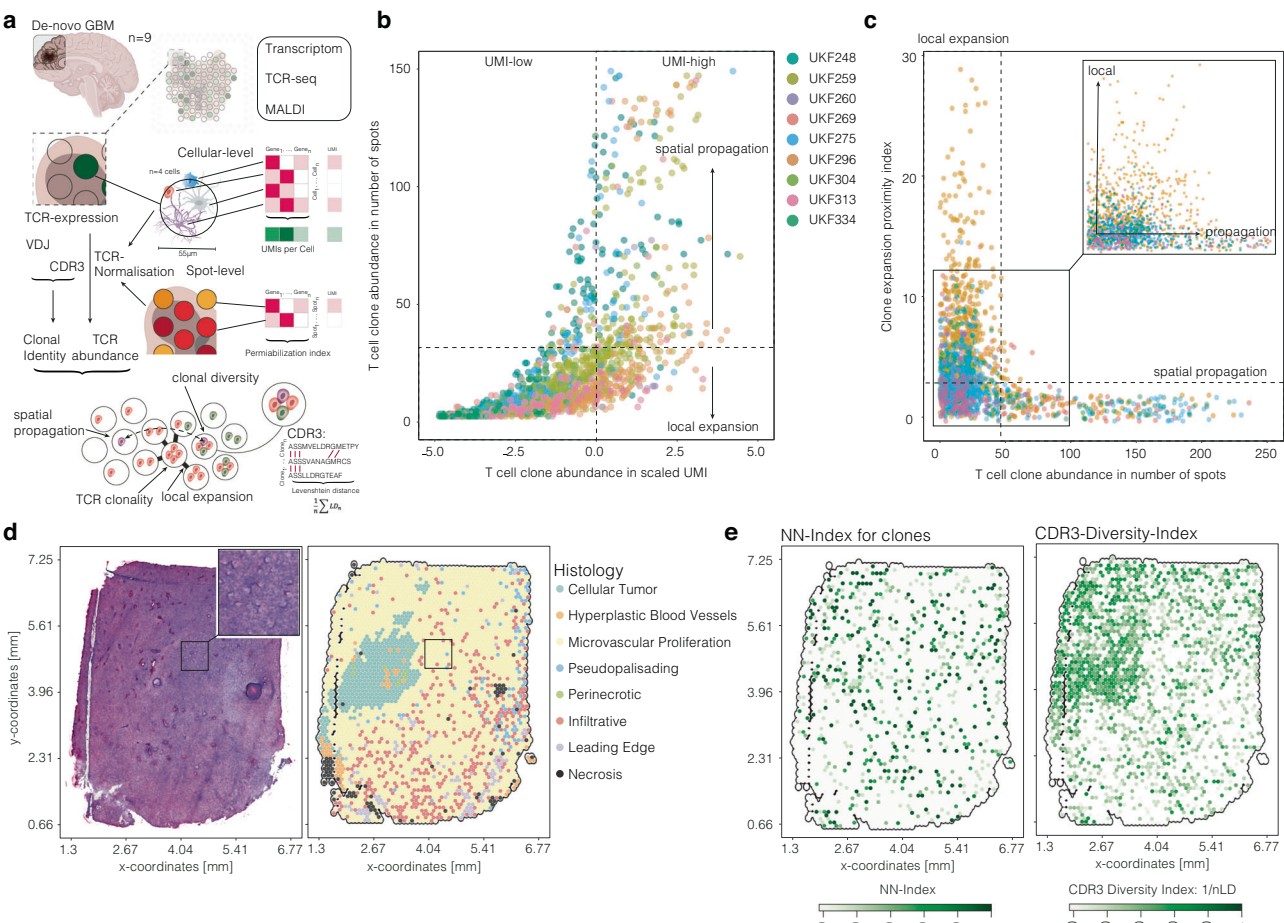

**Fig. 4 | Spatial distribution of T cell clones in glioblastomas. a** Illustration of analysis and postprocessing of spatial TCR. **b** Scatterplot representation of the T cell clone abundance (*x*) and spatial extension of the clone (by number of spots on the *y*-axis). Colors indicate the samples. **c** A scatterplot shows the neighborhood likelihood (*y*-axis) and the spatial extension of the clone (by number of spots on the *x*-axis). **d** Presentation of the H&E image and histological annotations (*x−y* axis indicates the space in mm). **e** Surface plot of the Clonality-Index (left) and the CDR3 amino acid diversity (right). Surface plots contain the dimension of the samples (in mm) on the *x* and *y* axes. Partially created with BioRender.com.

exhausted signatures, we identified that large parts of the "local T cell diversity" pattern showed enriched expression of exhausted genes, Supplementary Fig. 3. Some parts of the "local T cell diversity" along with the "local T cell expansion" pattern showed upregulation of classical cytotoxic markers suggesting that only a minority of the detected clones contribute to anti-tumor immune response.

## Clonal T cell expansion is associated with defined local cellular ecosystems

Given the spatial heterogeneity of T cell expansion and clonality, we aimed to investigate the differences in the local cellular ecosystem that might impact the two different T cell behaviors. To this end, we used two models: 1: Hierarchical cell type composition model (HCCM), 2: Node-centric expression modeling (NCEM[35]). In the first model, we aim to investigate the cellular topography of the different T cell expansion patterns and analyze the cellular communication required to reach defined cellular differentiation. Since the Visium data is limited by the 55 μm spot size, we preprocessed our data by single-cell deconvolution utilizing CytoSpace for the HCCM model and Cell2location for the NCEM model. We first analyzed the cellular composition of spots defined as local T cell expansion (defined by clones with high UMI counts >5 and clonality-index >1) and found three different patterns of cellular compositions. An NK-cell enriched niche, a T cell memory enriched niche, and a niche dominated by bone-derived macrophages previously defined by high expression of MHC class II molecules and B

cells, Fig. 6b. In a hierarchical layout, all three composition patterns are connected by interferon-gamma-activated CD4 T cells. The NK-rich niche was enriched for cytotoxic CD8 T cells and various stromal cells above all perivascular fibroblasts, also named "cancer-associated fibroblasts". In the myeloid-enriched niches, we frequently observed dendritic cells connected to B or proliferating T cells. To further understand the cell–cell communication, we employed the NCEM model, and separated regions with clonal T cell expansion based on their likelihood to express cytotoxic or exhausted/dysfunction genes derived from the cell type-specific gene expression matrices. The NCEM model displayed strong inputs from high MHCII expressing bone-derived macrophages and NK cells on CD4 T cells in regions with preferentially cytotoxic signatures. Overall, NK cells showed a dominating role in signaling required for local cytotoxic T cell expansion, Fig. 6c, d. In spots exhibiting increased expression of T cell dysfunction and exhaustion markers, the cell communication model revealed that, compared to the local clonal expansion pattern, there was increased input from MHC class II-expressing cells such as bone-derived macrophages, plasmacytoid dendritic cells (pDCs), and conventional dendritic cells. Bone-derived macrophages preferentially signaled not only to CD4 cells but also strongly to B cells, Fig. 6e, f. B cells, already identified as an important cell type in the hierarchical model, exhibited robust communication with natural killer (NK) cells. This B cell-NK cell axis was not observed in the cytotoxic regions. The role of B cells in the tumor microenvironment remains incompletely understood; however,

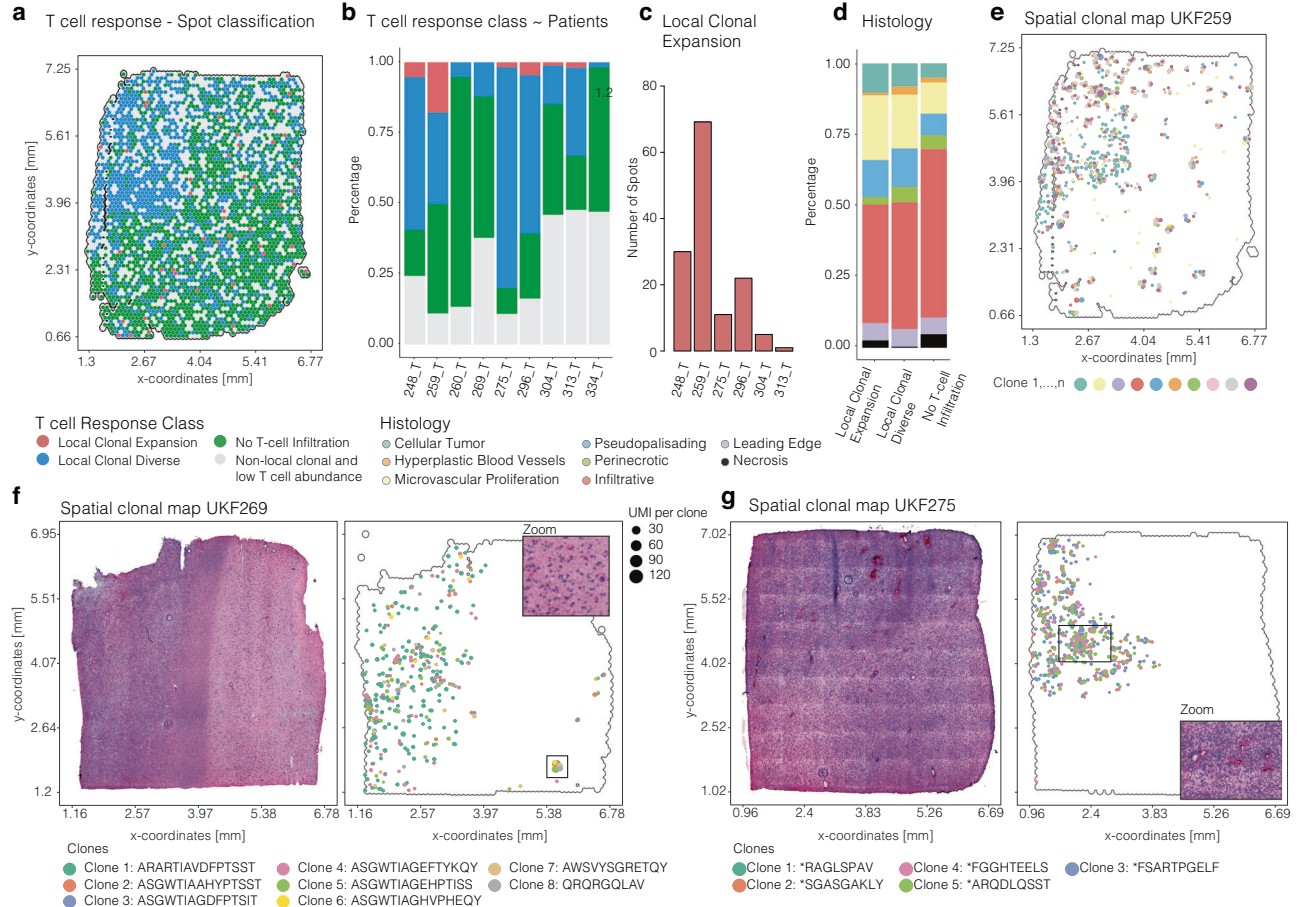

**Fig. 5 | Clonal annotation and single T cell clone visualization. a** Representative example of the sample UKF259. Colors indicate the TCR label of the spot. **b** Stacked barplot of the distribution of T cell response classes (indicated by colors) across samples. **c** Barplot of the number of spots classified as clonal. **d** Stacked barplot of the distribution of histological groups across T cell response classes. **e–g** Surface plots with histology (left) and spatial mapping of single clones. Surface plots contain the dimensions of the samples (in mm) on the *x* and *y* axes.

a growing body of literature has attributed B cells to a highly immunosuppressive role in glioblastoma, which aligns with our findings[36].

## Extracellular matrix genes and Platelet Factor 4 shape the T cell response in distinct niches

Comprehending the various signaling mechanisms in the tumor microenvironment that influence T cell behavior towards anti-tumor immunity or exhaustion is of paramount significance. Our findings indicate that the primary factors affecting differential T-cell behavior involve the B-cell-NK axis (dysfunctional/exhaustion) and the NK-CD4 axis (cytotoxic). To further delineate the specific signaling perturbations and conditions under which they occur, we examined receptor-ligand interactions using cell type-specific gene matrices, as shown in Fig. 7a. We began by investigating regional cell–cell signaling, with B cells as the sender and NK cells as the receiver. In addition to classical immune-related receptor-ligand signaling molecules, we identified various extracellular matrix genes, particularly Collagen type VI alpha 1 (*COL6A1*), and its receptor on NK cells, Syndecan-4 (*SDC4*). *SDC4*, a cell surface heparan sulfate proteoglycan involved in cell adhesion, migration, and signaling, exhibited significant enrichment in niches with increased T-cell exhaustion, Fig. 7b–d. The spatial proximity of these genes' expression was visualized in a co-expression surface plot, revealing a distinct local pattern, Fig. 7e. Beyond this cell–cell interactions, we also found known immunosuppressive interactions such as *ACVR1-BMP7* which is a type I receptor for the transforming growth factor-beta (TGF-β) superfamily, which plays essential role in T cell

regulation and exhaustion[36]. In our previous analysis, we determined that increased signaling from NK cells to CD4 cells correlated with enhanced T cell response. We used the same approach to examine the NK (sender) and interferon-gamma-driven CD4 (receiver) axis, Fig. 7f, g. We identified signaling between Platelet Factor 4 (*PF4*), a small chemokine primarily secreted by activated platelets in response to injury or inflammation, and C-X-C chemokine receptor type 3 (*CXCR3*). Annotating this receptor-ligand interaction across cytotoxic or dysfunctional regions confirmed spatial accumulation in areas associated with cytotoxic T-cell response, Fig. 7h, i. Our findings highlight the importance of understanding signaling mechanisms in the tumor microenvironment that influence T-cell behavior, specifically the B-cell-NK axis driving exhaustion and the NK-CD4 axis promoting cytotoxic responses. We identified key receptor-ligand interactions, such as *COL6A1-SDC4* and *PF4-CXCR3*, spatially linked to distinct T-cell behaviors.

## T cell exhaustion is associated with a metabolic switch towards enhanced glycolysis

Our data suggests that local T cell expansion predominantly comprises phenotypically exhausted T cells, driven by immunosuppressive signaling within the tumor microenvironment and other lymphoid cells. Defining T cell exhaustion using only TCR-seq and transcriptional data is challenging due to functionally distinct T cells (CD4/CD8/TREG/NK) bearing a TCR; therefore, we integrated an additional molecular layer of metabolomics to functionally characterize identified T cell clones. In

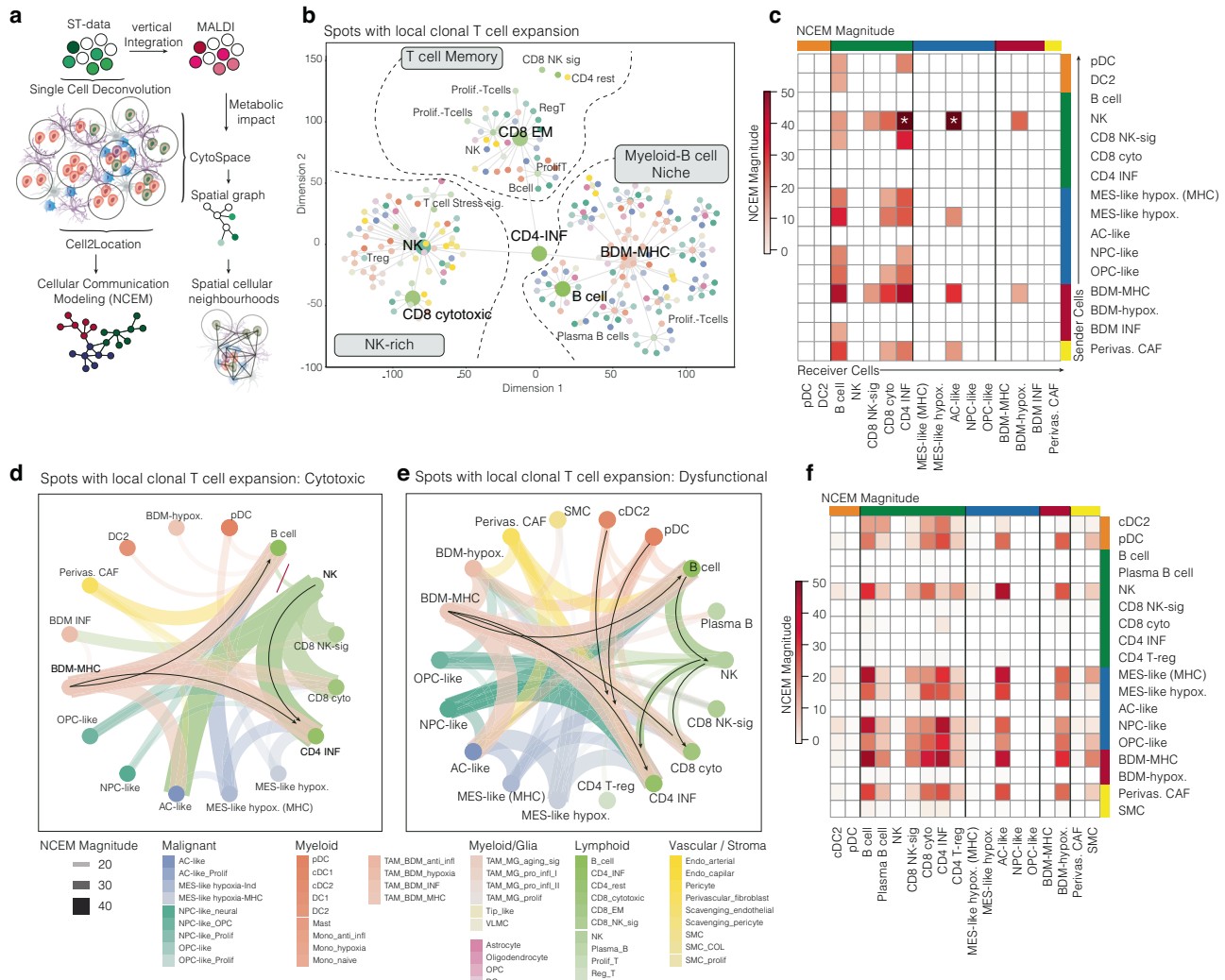

**Fig. 6 | Neighborhood analysis of exhausted and cytotoxic T cell clones. a** Illustration of the spatial cell communication modeling workflow. **b** Hierarchical graph model of the likelihood of spatial cellular co-existence in spots with local T cell expansion. Cells are presented as nodes and the size represents hub cells in the network. Edges indicate the likelihood of cellular proximity. **c** Heatmap of the NCEM with sender cells (rows) and receiver cells (cols) in cytotoxic regions.

**d**, **e** Circle plot of the NCEM graph for cytotoxic areas (left) and dysfunctional/ exhausted areas (right). Colors indicate the cell types. The size of the edges indicates the magnitude of the cell-cell communication. **f** Heatmap of the NCEM with sender cells (rows) and receiver cells (cols) in exhausted regions. Surface plots contain the dimensions of the samples (in mm) on the *x* and *y* axes. Partially created with BioRender.com.

cancer, T cell exhaustion is characterized by metabolic alterations, transitioning from oxidative phosphorylation (OxPhos) in cytotoxic T cells to increased glycolysis in exhausted T cells. This metabolic switch can serve as a predictive marker for T cell behavior. To this end, we added an additional molecular layer of spatially resolved metabolomics (MALDI, $n = 5$) as recently described[4], to functionally characterize identified T cell clones. Integrated profiles containing gene expression and metabolic abundance data were computed, extracting clone-specific multi-omic profiles, Fig. 8a. Spatial correlation analysis with T cell-specific expression of cytotoxic or exhausted marker genes revealed significant enrichment of hypoxia-associated metabolic pathways (e.g., gluconeogenesis, glycolysis, glutathione metabolism) in exhausted T cells, Fig. 8b. Single clone analysis demonstrated that clones with increased transcriptional exhaustion signatures were significantly enriched for glycolytic metabolites ($R^2 = 0.534$, $p < 2.2 \times 10^{-16}$), correlating with increased hypoxic conditions in regions with spatial accumulation of exhausted clones, Fig. 8c-d. These findings align with our initial investigation of enriched T cell clones in perinecrotic regions and microvascular proliferations, Fig. 2f. In summary, we observed increased mesenchymal-like tumor cell abundance,

bone-derived macrophages, CD4 cells, and B cells in hypoxia-associated regions, implicated in enhanced genomic instability and mutational burden[4]. This potential driver of clonal T cell expansion is closely linked to immunosuppressive environments promoting T cell dysfunction and exhaustion. The regional metabolic situation, as well as the T cell shift towards glycolytic metabolism, exacerbate T cell dysfunction. T cells can only locally expand in regions less affected by this detrimental cycle; however, the scarcity of immunosuppression released spots with effective antigen presentation constrain their antitumor efficacy.

## Discussion

In this study, we utilized spatially resolved T cell receptor sequencing (SPTCR-seq) to investigate T cell responses in GBM, a "cold" tumor with overall low T cell abundance. Through target enrichment, long-read sequencing, and read consensus correction, we gained insights into the spatial heterogeneously distributed T cell response. In all available spatial transcriptomic methods, the reconstruction of TCRs necessitates the enrichment of TCR-containing fragments and amplification for TCR-seq. To date, three methods, including our presented

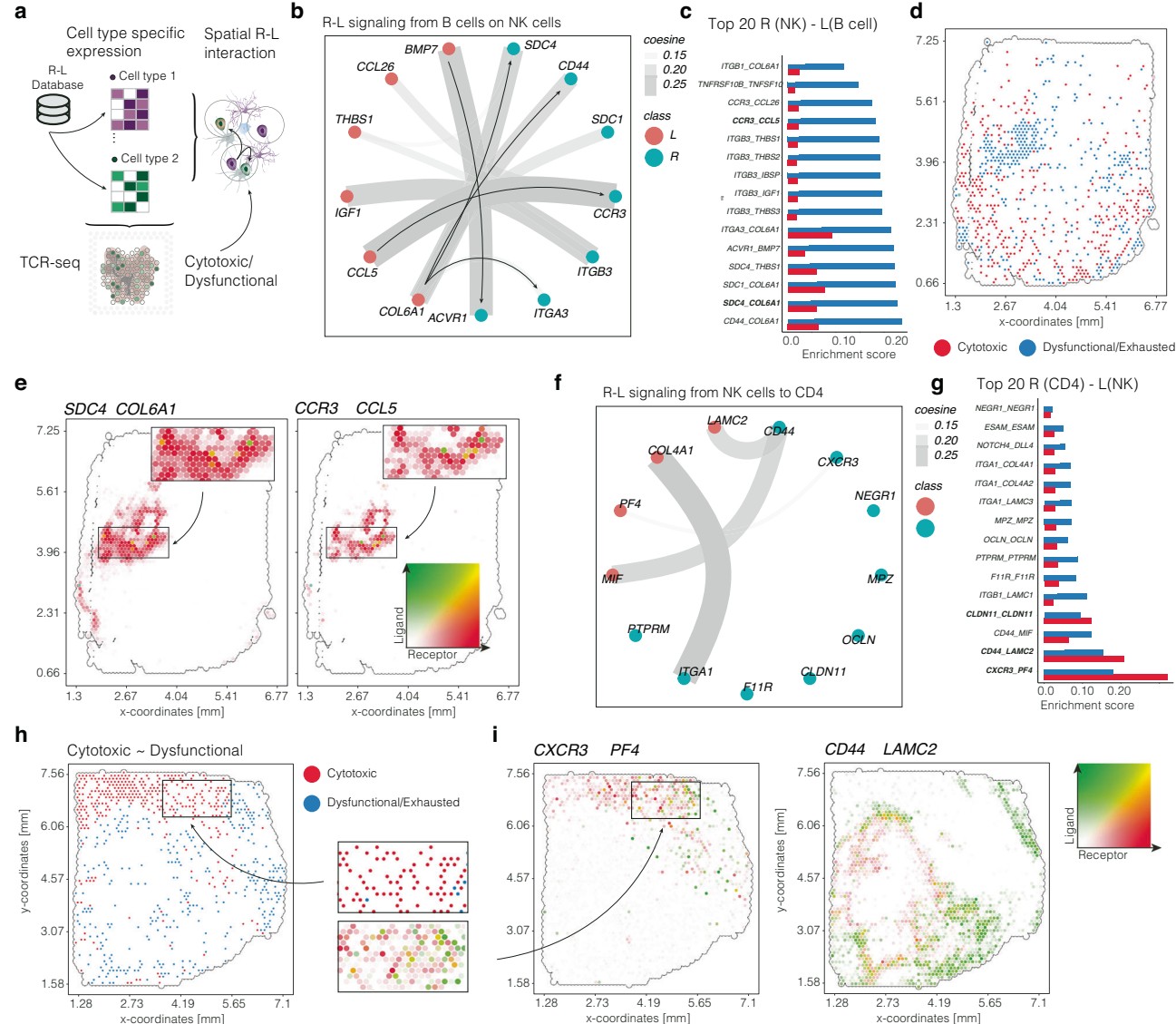

**Fig. 7 | Receptor Ligand Interaction at spatial resolution. a** Illustration of the workflow for spatial R-L communication modeling using cell-type specific metrices. **b** Circle plot of the most abundant R−L interaction from B cells towards NK cells. **c** Barplot of regional annotation of cytotoxic/exhausted regions compared to the high-scoring R-L pairs. **d** Example surface plot of the cytotoxic (red) or exhausted (blue) regions in sample UKF 259. **e** Mutual expression plot using RGB colors (from green to red). The color indicates the level of overlap between the ligand and receptor. **f** Circular plot of the R-L graph between NK cells as sender and CD4 (IFN) as receiver cell. **g** Barplot of regional annotation of cytotoxic/exhausted regions compared to the high-scoring R-L pairs. **h, i** Surface plot of the cytotoxic/exhausted regions and specific R−L interactions. Surface plots contain the dimensions of the samples (in mm) on the *x* and *y* axes. Partially created with BioRender.com.

approach, have been published for performing TCR-sequencing in spatial transcriptomics. We conducted a comparative analysis of these distinct methodologies, identifying significant disparities in their capacity to fully annotate TCRs. SPTCR, which employs targeted enrichment through hybridization, demonstrated superior performance than other PCR-based methods, owing to its elevated success rate in TCR reconstruction.

Additionally, SPTCR-seq effectively minimizes PCR amplification cycles, identified as a major source of sequencing errors in our analysis. In contrast to the alternative methods, Hudson's (70 [35 + 35] PCR cycles) and Liu's (30 [18 + 12] PCR cycles), SPTCR-seq necessitates a total of only 23 (5 + 18) PCR cycles. This reduction in PCR cycles, coupled with consensus-based error correction, enhances sequence accuracy and contributes to more sensitive and specific TCR detection. We acknowledge that our comparative analysis was primarily influenced by the specific sequencing protocols utilized, particularly the exclusive use of long-read sequencing for our SPTCR-seq method.

Future studies should explore the potential of adapting multiplex-PCR methods to long-read sequencing, as this could significantly shift the performance landscape of various spatial-TCR methods.

We leveraged SPTCR-seq to explore spatially diverse T cell response and showed exhausted T cell response regions were dominated by anti-inflammatory tumor-associated macrophages, regulatory T cells, and tumor cells. In contrast, reactive T cell phenotype regions displayed stronger signals from dendritic cells and NK cells. However, these islands of immune response were small, as the tumor exploited immune cells to attract T cells into areas with high macrophage and tumor cell densities, leading to T cell exhaustion. We demonstrated the power of spatially resolved technologies in extracting defined cellular interaction of the tumor microenvironment which have impact on T cell response. Our findings are significantly constrained by the technology, which relies on spatial proximity but does not elucidate the functional aspects of cellular interactions. The absence of a comprehensive evaluation of our reported results

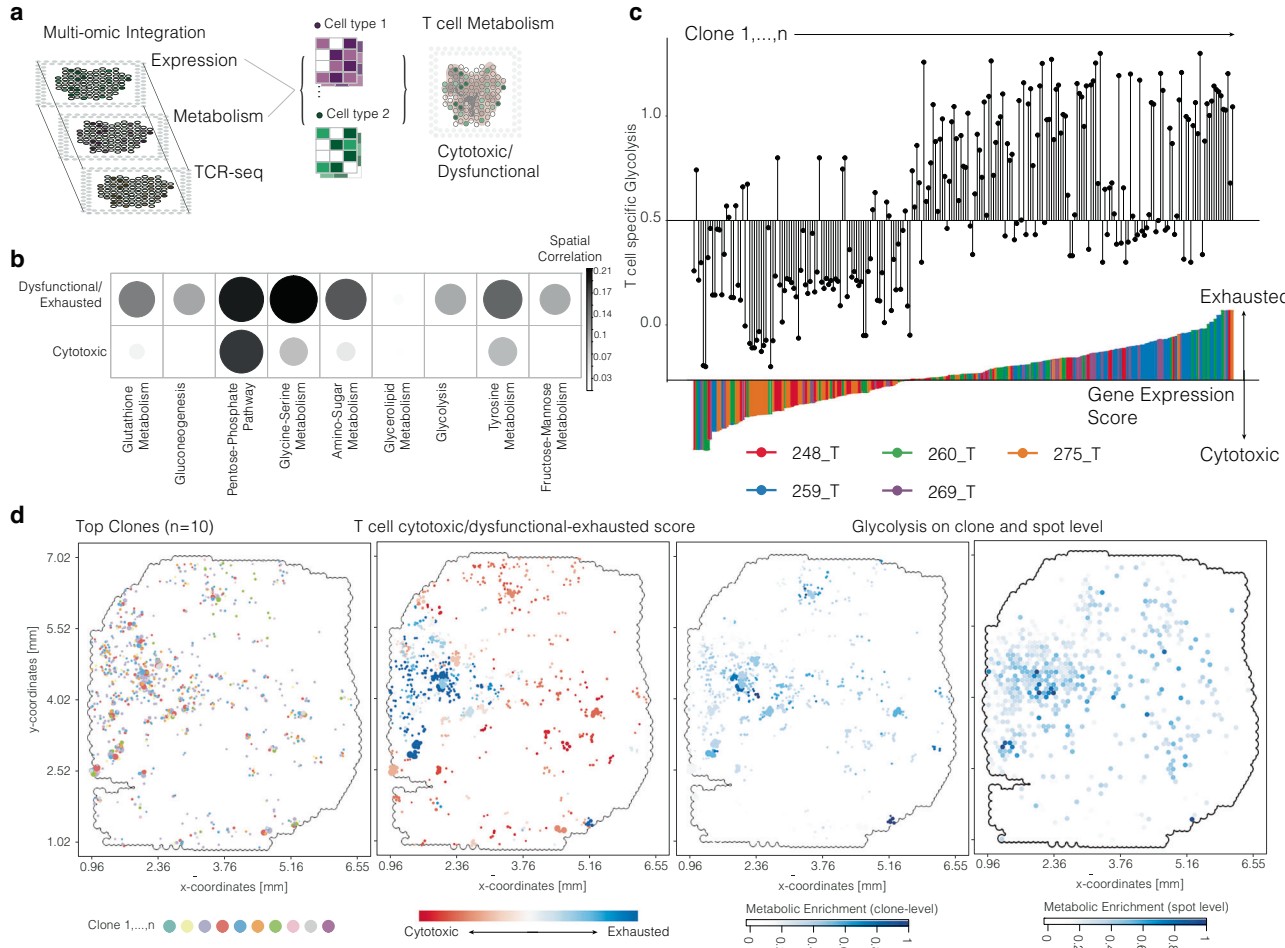

**Fig. 8 | Multi-omic annotation of T cell metabolism. a** Illustration of the workflow for integration of metabolic and transcriptional data. **b** Dotplot for correlation between transcriptionally defined T cell exhaustion/cytotoxic behavior and abundance of metabolites from various pathways. **c** Illustration of glycolysis (top) and the exhaustion/cytotoxic score (bottom) at clone level. All spots occupied by a clone are summarized for the clonal profile. **d** Surface plots of an example sample with the abundance of clones (from left to right), the exhaustion/cytotoxic score, and the abundance of glycolysis metabolites on clone- and spot-level (right). Surface plots contain the dimensions of the samples (in mm) on the x and y axes.

considerably restricts the generalizability of our observations. Although SPTCR-seq exhibits higher sensitivity compared to previous studies, several limitations persist, including sample size, sampling bias, and the inherent heterogeneity of GBM, which further curtail the applicability of our reported findings. Moreover, spatial transcriptomics data is inherently noisy and sparse, with interpatient variability potentially influencing our findings. Investigating larger and more diverse patient cohorts along with functional characterization is essential for addressing variability and translating therapeutic targets into applications. Lastly, our study does not include clinical outcome data, such as patient survival or therapy response, which would be valuable for a more comprehensive understanding.

To improve accessibility and applicability, we developed an optimized pipeline for long-read processing and error correction, available on our GitHub repository and attached a user-friendly step-by-step protocol for the target enrichment workflow. This pipeline integrates with existing Visium workflows and the SPATA framework using the SPATAImmune R-based package. The combination of efficient workflows and user-friendly software will support the widespread use of SPTCR-seq, with anticipated technical and computational improvements accelerating its adoption across the scientific community, providing a tool to explore spatial heterogeneity of T cell response.

In future developments, our protocol could be adapted for advanced applications by making minor modifications to the enrichment library, specifically the target probes. Such adaptability allows for enhanced annotation through spatially resolved B cell receptor sequencing or high-coverage, full-transcriptome sequencing. These advanced capabilities could facilitate the detection of splice variants or mutations.

## Methods

### Ethical statement

The study design, data evaluation, and imaging procedures were given clearance by the ethics committee at the University of Freiburg, as delineated in protocols 100020/09 and 472/15_160880. All methodologies were executed in compliance with the guidelines approved by the committee. Informed consent, in written form, was received from all participating subjects. The Department of Neurosurgery of the Medical Center at the University of Freiburg, Germany, was responsible for securing preoperative informed consent from all patients participating in the study. Supplementary Data 1 provides a summary of the clinical characteristics of the patients.

### Spatial transcriptomics data and availability

Spatial transcriptomics experiments were performed as recently described using the 10X™ Visium Spatial Gene Expression kit

(https://www.10xgenomics.com/products/spatial-gene-expression).
Tissue optimization, RNA permeabilization and Illumina library preparation working steps were performed according to the manufacturer's instructions. For SPTCR-seq, we used 10X Visium full-length cDNA libraries from 9 primary glioblastoma samples, acquired in a recent in-house spatial transcriptomics study[4] and described in detail in Supplementary Data 1.

## Probe design for tissue collection and RNA quality control

Fresh tissue samples were collected immediately after resection and snap-frozen in isopentane after embedding in Tissue-Tek O.C.T. Compound. The embedded tissue was stored at −80 °C until further processing. RNA integrity was determined, and samples with an RNA integrity number greater than 7 were used for subsequent analysis.

## SPTCR-seq target enrichment probe design

In this study, we designed probes for target enrichment by identifying functional T-cell receptor sequences located within the TRA, TRB, TRD, and TRG loci of the human genome (hg38/GRCh38) using the IMGT database We generated 100nt-sized probes using the KAPA™ Hyperdesign™ online tool (https://www.hyperdesign.com) and had them manufactured by Roche®. To ensure high specificity, we allowed a maximum of 20 matches to the human genome for each probe. We set a default value of 100 bp for the probe window placement on genomic segments to achieve comprehensive coverage of T-cell receptor segments. Our design covered a total of 186 regions, utilizing 211 unique capture probes. A list of genes, genomic coordinates, and ENSEMBL gene codes used in the probe design is available in Supplementary Data 2. The SPTCR-seq workflow includes a target enrichment process consisting of four major steps. The full-length cDNA is generated after reverse transcription and second strand synthesis, as described in the 10X Visium fresh tissue protocol (CG000239 Rev C). The following sections outline the SPTCR-seq workflow, Supplementary Information.

**Step 1: Pre-capture amplification**. Before target enrichment, the library underwent pre-capture amplification using 100 ng of unfragmented cDNA, dissolved in 20 μl of nuclease-free water. A primer mix (consisting of 1.5 μl TSO (100 μM), 1.5 μl R1 (100 μM), and 2 μl nuclease-free water) was added to the PCR mix. The final volume was brought to 50 μl by adding 25 μl of KAPA HIFI HotStart Ready-mix (Roche Kapa Biosystems). After thorough mixing, PCR was performed under the following conditions: Step 1: 98 °C for 3 min, followed by five cycles of {Step 2: 98 °C for 20 s, Step 3: 65 °C for 30 s, Step 4: 72 °C for 2 min}, and ending with 72 °C for 3 min and 4 °C hold. Following PCR, a SPRI bead cleanup and size selection were performed to select transcripts greater than ~150 bp, using Roche Kapa HyperPure Beads according to the manufacturer's instructions for NGS workflows. Finally, the library was quantified using a Thermo Fisher Qubit system (1X high-sensitivity dsDNA Kit), and the size distribution was measured using an Agilent 5300 Fragment Analyzer system operated with the Agilent DNF-920 Reagent Kit to ensure proper amplification of the library.

**Step 2: Target enrichment probe hybridization**. In this study, we used 1.5 μg of the amplified, unmultiplexed cDNA library and filled it with PCR-grade water to a final volume of 45 μl. To block repetitive regions in the sample and increase the on-target rate of enrichment, we added COT-Human DNA (included in Roche HyperCapture Reagent Kit) and Blocking Oligos (Partial R1: 5′-CTACACGACGCTCTTCCGATCT-3′, Partial TSO: 5′-AAGCAGTGGTATCAACGCAGAG-3′, ordered from Thermofisher). We performed a SPRI-bead cleanup using 130 μl of KAPA HyperPure beads. The bound cDNA was eluted from the beads by sequentially pipetting the following agents directly above the magnet-bound, dried beads: 2.5 μl of TSO (100 μM), 2.5 μl of Read 1 (100 μM), and 8.4 μl of PCR-grade H2O. The resulting bead suspension (13.4 μl)

was mixed thoroughly. We then combined 28 μl of Roche hybridization buffer, 12 μl of Roche hybridization component H, and 3 μl of PCR-grade water in a hybridization mix and added it to the bead suspension. Finally, 56.4 μl of the eluate was mixed with 4 μl of the KAPA target enrichment probes, prepared according to the manufacturer's instructions. After double-strand denaturation at 95 °C for 5 min, the solution was incubated for 16 h at 55 °C, maintaining the samples at 55 °C throughout the remaining steps.

**Step 3: Wash and recover**. Prepare wash buffers according to the manufacturer's protocol and KAPA Hypercap Workflow V3.0. After the 16-h incubation with target enrichment probes, mix the sample with prepared capture beads and incubate for 15 min. Wash the capture beads and elute the bead-bound DNA using wash buffers with decreasing pH and salt concentration, following the manufacturer's instructions for KAPA HyperCap. Finally, suspend the DNA in 20 μl of PCR-grade water.

**Step 4: On-target amplification**. Next, we amplified the bead-bound target-enriched T-cell receptor transcripts with a final round of PCR amplification using 2.5 μl 10X primers TSO and Read 1 each at 20 μM, along with 25 μl Kapa HiFi Hotstart ReadyMix, and 20 μl of the sample. The PCR conditions were: Step 1: 98 °C for 3 min, Steps 2-4: (98 °C for 20 s, 65 °C for 15 s, 72 °C for 1:30 min) × 18 cycles, Step 5: 72 °C for 3 min, Step 6: 4 °C HOLD. Perform a SPRI bead cleanup with 70 μl KAPA HyperPure Beads according to the manufacturer's instructions. Suspend the final purified result in 22 μl Tris-HCl, using 2 μl for Qubit quantification and size distribution measurement with the Agilent Fragment Analyzer operated with the Agilent DNF-920 Reagent Kit. Store the remaining sample at −20 °C for several months. An example of the library is provided in Supplementary Fig. 4.

## Nanopore sequencing

To prepare the enriched cDNA for sequencing, we used ONT 1D sequencing. For samples #UKF260GBM, #UKF334GBM, #UKF248GBM, #UKF275GBM, #UKF304Recurrent, #UKF269GBM, #UKF313GBM, and #UKF296GBM, we utilized the sequencing by ligation kit (SQK-LSK110) and sequenced each sample individually on single flow cells (FLO-MIN106D). The SPTCR and Visium Sample ID, as well as flow cell ID, can be found in Supplementary Data 2. We used 200-300fmol as input for the library preparation with R9.4 Chemistry. The library preparation workflow for samples #UKF260GBM, #UKF334GBM, #UKF248GBM, #UKF275GBM, #UKF304Recurrent, #UKF269GBM, #UKF313GBM, and #UKF296GBM was performed according to the manufacturer's protocol for SQK-LSK110 (GDE_9108_v110_revE_10Nov2020.pdf), using an input of 200-300fmol. We opted to enrich for fragments <3 kb by using adapter ligation and the provided Short Fragment Buffer. For theresequencing of #UKF275GBM, #UKF296GBM we employed FLO-MIN114 with R10.4.1 flow cells and the Library Preparation for Sequencing by Ligation V14 chemistry. In the final loading step, we used ~5 fmol of the sequencing library and sequenced samples individually on single-flow cells. Base calling for the samples sequenced on R9.4 was performed offline using Guppy GPU (v4.0) on a Nvidia GeForce 2080ti in super-high accuracy mode with the following parameters:

```
guppy_basecaller
-i 'path/to/fast5/folder'
-s 'path/to/output/folder
-x cuda:0
-c dna_r9.4.1_450bps_sup.cfg
-calib_detect
--gpu_runners_per_device 16
--num_callers 8.
```

For R10, V14 chemistry samples, we performed base calling using the PyTorch-based base caller of Oxford Nanopore Technologies, Bonito (https://github.com/nanoporetech/bonito) with the following command:

```
bonito basecaller dna_r10.4.1_e8.2.260bps_sup@3.5.2
'path/to/fast5/folder'
--device "cuda"
--chunksize 4000
--min-qscore 10
--overlap 1000
--batchsize 1000
> path/to/output.fastq
```

A list of samples, chemistries, flow cell identification numbers as well as detailed concentrations for checkpoints can be found in Supplementary Data 2.

### rhTCR Seq for Visium samples

To compare our SPTCR-seq method with the recently developed Slide TCR-seq protocol[13], we tailored the protocol to align with the specific properties of the 10X Visium technique. As delineated in the rhTCR-seq protocol by ref. [13] and the Slide-TCR-seq protocol in ref. [14], we prepared libraries for the RNase H-dependent PCR (rhPCR) reaction. First, we prepared a stock of 20x rhPCR buffer formulated with a final concentration of 300 mM Tris-HCl (pH 8.4), 500 mM KCl, and 80 mM MgCl2. RNase H2 was diluted to 20 mU/mL utilizing RNase H2 Dilution Buffer (IDT). For human TCRs, we ordered 69 rhPCR primers specific for the V segments of human alpha and beta TCR genes (Rd2.AV.x1/Rd2.BV.x1) from IDT. These primers were pooled at a concentration of 5 mM each, by combining 5 μL of each primer at 500 μM with 155 μL of TE buffer. To prepare the TCR amplification and indexing PCR reaction, we mixed 6 μL Visium cDNA library (0.5 ng/μL) with 8 μL of Index Primers from one well of the 10X Index Kit TT Set A. Subsequently, we incorporated 10 μL of rhPCR master mix (final concentration: 1x rhPCR Buffer, 400 μM dNTPs, 50 nM of Rd2.AV.x1 and Rd2.BV.x1 primers, 0.5 mU/mL RNase H2, and 0.2 units/mL OneTaq hot-start DNA polymerase [New England BioLabs M0481L]). We executed the following program on the thermal cycler: one cycle of 95 °C for 5 min, 18 cycles of 96 °C for 20 s, 60 °C for 4 min, and 72 °C for 2 min, and subsequently held at 4 °C.

Upon completion of the PCR reaction, we purified the product using 12 μL (0.5×) Roche HyperPure Beads, quantified it with a Qubit Fluorometer, and assessed its quality employing an Agilent Fragment Analyzer 5300. Uniquely indexed libraries were then pooled at a concentration of 2 nM and sequenced on an Illumina NextSeq Sequencer using NextSeq 1000/2000 P2 Reagents (300 Cycles) v3 Flowcells at the Neuropathology UK Freiburg facilities.

### Comparison to Visium TCR protocol by Hudson et al.

To implement the TCR sequencing method delineated in Hudson et al., we ordered the TRBV-specific primer pool from IDT as specified in the publication and pooled it in accordance with the outlined protocol[12]. Subsequent to pooling, 5 μl of amplified Visium cDNA was utilized as a template in a 35-cycle PCR reaction, employing the pooled TRBV forward primers, 10X Amplification Mix, and 10 μM Partial Read 1 primer. The resulting PCR product was purified without fragmentation using SPRIselect beads and quantified with a Qubit 1X dsDNA HS Assay Kit (Thermo Fisher Scientific). Sample index PCR was performed using primers from the 10X Genomics Dual Index Kit TT Set A and Amplification mix from the 10x Genomics Library Construction Kit, following the manufacturer's guidelines (protocol CG000239, 10X Genomics). Upon bead purification, libraries were sequenced on an Illumina NextSeq instrument utilizing NextSeq 1000/2000 P2 Reagents (300 Cycles) v3 at the Neuropathology UK Freiburg facilities.

### MixCR TCR reconstruction for illumina-based TCR-seq protocols

To retain the barcode and UMI Information of the fastq header, we applied umi_tools extract --bc-pattern=CCCCCCCCCCCCCCCCNNNNNNNNNNNN on the fastqs, specifying read1 and read2. Then to reconstruct TCRs from short reads, we utilized MixCR, a robust tool for T-cell receptor (TCR) repertoire analysis. We executed a pipeline designed aligned with the exemplary code for slideTCR-seq provided in https://github.com/soph-liu/Slide-TCR-seq/blob/main/mixcr_code.txt as template. The implemented pipeline consisted of several steps, including alignment, assembly, contig assembly, and export of clones and alignments. Initially, the pipeline performed alignment of input reads against reference V, D, and J genes, generating an intermediate vdjca file. Next, the alignments were assembled into clonotypes, producing a clna file. The contig assembly step further refined the clonotypes and generated a.clns file. Finally, the pipeline exported the assembled clonotypes and alignments, creating two output files: one containing the detailed information on clones and another including the alignments associated with clone IDs. Next, we used mixcr exportClones to generate the clones.txt file and the command mixcr exportReadsForClones to extract the used reads as fastqs for the clones.

```
mixcr align -Xmx50g -r ${SAMPLE_NAME}.report -s hsa -Osa
veOriginalReads=true        ${OUTFOLDER}/${SAMPLE_NAME}
_extracted_R1.fastq.gz      ${OUTFOLDER}/${SAMPLE_NAME}
_extracted_R2.fastq.gz ${SAMPLE_NAME}.vdjca -f
mixcr assemble --write-alignments ${SAMPLE_NAME}.vdjca
${SAMPLE_NAME}.clna -f
mixcr  assembleContigs  ${SAMPLE_NAME}.clna  ${SAMPLE_
NAME}.clns -f
mixcr exportClones -cloneId -targets -f -p fullImputed
-topChains ${SAMPLE_NAME}.clns ${SAMPLE_NAME}_clones.
txt
mixcr exportAlignments -f -descrsR2 -descrsR1 -cloneId
-readIds  -targetSequences  -topChains  -chains  -clo
neIdWithMappingType ${SAMPLE_NAME}.clna ${SAMPLE_NAME}
_cloneID.txt
```

### SPTCR computational pipeline

**Demultiplexing and UMI correction.** For demultiplexing, we employed scTagger[26], which uses a direct matching approach for barcodes. We matched Visium dataset tissue barcodes from the 10X Space Ranger output to the long reads, allowing for up to two mismatches as per scTagger's default string edit distance values. To extract the UMI region of the read, we identified the Illumina Adapter Read1 (CTACACGACGCTCTTCCGATCT) sequence as an anchor. We conducted two scTagger runs: one to extract the first 16 bp after the adapter match for barcode matching, and another to extract 28 bp after the match, considering the last 12 bases as UMIs for subsequent UMI correction. The detected barcodes and extracted UMI regions were tabulated, and unprocessed reads were used in downstream correction pipeline steps. To correct amplification errors, we utilized directional UMI clustering with the UMI-Tools suite. This enabled accurate quantification of TCR sequences in spatial transcriptomics experiments by correcting counts to the UMI-cluster count per T-cell receptor.

**Splitting, reorientation, and extraction of transcript inserts.** To orient the inherently undirected reads generated by ONT, we employed the Pychopper package from Oxford Nanopore Technologies (https://github.com/epi2me-labs/pychopper). This algorithm restores the original 5′−3′ direction of the reads and splits fused reads by matching the provided 10X adapters Read1 (R1) and the template switch oligo (TSO)

to the long read. After splitting and reorienting the reads, we used Cutadapt[28] to trim synthetic regions introduced during the library preparation of 10X and ONT (https://github.com/marcelm/cutadapt). The extracted insert was then utilized for the remaining bioinformatics workflow.

**Annotation of target-enriched TCR reads and grouping by variable, joining, and constant segments.** In this method, target-enriched TCR reads were annotated and grouped based on variable, joining, and constant segments. The extracted inserts were subsequently used for primary annotation with PyIR (https://github.com/crowelab/PyIR), a Python implementation of the IgBLAST algorithm optimized for handling large files. We evaluated all currently available immune receptor annotation algorithms for their performance on ONT long reads (Supplementary Fig. 2), and IgBLAST demonstrated the best performance for annotating error-prone, contiguous immune receptor sequences. We selected the number of mappable and fully annotatable TCR reads and detected receptor segments as quality criteria. The annotated output was ultimately used to split the FASTQ files based on the detected constant and TCR segments. We then divided the FASTQs into fractions based on the same variable and constant segments, discarding off-target reads or reads without a detected variable segment. Finally, we saved each V, C FASTQ file to disk and utilized them as input FASTQs for the error correction process.

**SPTCR error-correction and data preprocessing.** Each V, C grouped FASTQ file was used as input for the RATTLE algorithm, clustering reads based on similarity[29]. Partitioning the FASTQs into multiple variable and constant segment clusters helped reduce memory footprint while improving read classification during consensus correction. A consensus was built for each cluster using SIMD partial order alignment (SPOA) (https://github.com/rvaser/spoa). The corrected and V, J, C-split FASTQs were merged and subsequently analyzed with IGBLAST. To evaluate the success of the correction, we aligned the FASTQs to the IMGT known constant T cell receptor regions using minimap2[50] and measured mismatches in SAM alignments before and after correction (Fig. 2c). Only constant segments were assessed for correction success, as V-, D-, J-recombination alters the variable, joining, and diverse T cell receptor segments in the germline.

### Data import for spatial data analysis
The cell ranger output was imported into SPATA2 using the import function (*SPATA2::initiateSpataObject_10X*). The import function also perform baseline sample processing using the recent described pipeline[4]. Our package SPATAImmune:: imports T cell receptor sequences (SPTCR pipeline) into the SPATA object on the slot *@data*. Import of the SPTCR outputs can be either performed from the csv files (*…/{sample}_CORRECTED_umi_corrected_count_table.csv*) or by providing the output path.

### SPTCR-seq preprocessing in SPATA
The provided functions are designed to process SPTCR-seq data through a series of preprocessing and normalization steps. These functions can be grouped into the following main steps:

preprocessTCR: This function filters and preprocesses the raw TCR data based on the given criteria, such as minimum and maximum CDR3 amino acid length, minimum unique molecular identifier (UMI) expression, and required annotation of the V, D, and J region.

preprocessLD: This function clusters similar TCRs based on the Levenshtein distance of their CDR3 amino acid sequences. For each CDR3 sequence (with unique UMI) $x_i$ in which $i$ represents the total

number of CDR3 amino acid sequences $\{x_1, x_2, \ldots, x_i\}$, we computed the similarity of a target sequence $y$ by

$$lev(x,y) = \begin{cases} |x| \text{ if } |y| = 0, |y| \text{ if } |x| = 0 \\ lev(tail(a), tail(b)) \text{ if } x[0] = y[0] \\ 1 + \min \begin{cases} lev(tail(a), b) \\ lev(a, tail(b)) \\ lev(tail(a), tail(b)) \end{cases} \end{cases} \quad (1)$$

TCRs were summarized if a specified minimum distance threshold (minimal distance = 2). Further, we grouped TCRs with similar CDR3 amino acid sequences, to calculate the mean identity values for V, D, and J regions, and selected the CDR3 amino acid sequences with the longest length as the representative sequence for the group. The number of unique TCRs before and after the clustering process is displayed, along with the mean number of spots per TCR.

normalizeTCR: This function normalizes the TCR data based on the UMI abundance and cellular density at each spot. The total UMI count and the number of cells per spot are retrieved from the input object and used to calculate the UMI count per cell for each spot. The TCR data is then filtered based on a minimum expression threshold and joined with the total UMI data. The TCR data are grouped by CDR3 amino acid sequence, and summary statistics (e.g., total reads, and number of spots) are calculated for each group. In summary, these functions preprocess, cluster, and normalize SPTCR-seq data to identify and analyze TCR clones within a spatial transcriptomics context. The processed data can then be used to investigate clonal relationships, diversity, and spatial patterns of TCRs.

### Visualization of T cell clones
We utilized a robust approach to visualize clones (variable *c*) in space by analyzing the top $\{c_1, c_2, \ldots, c_n\}$ most frequently occurring clones (variable n) within a given sample. To achieve this, we first selected the target clones by arranging them in descending order based on the number of spots and then randomly sampling n clones for further analysis. The expression data of these clones was obtained by aggregating the spot-wise TCR (T-cell receptor) data for each selected clone and computing the mean expression. Next, we integrated the spatial coordinates of each spot by joining the TCR expression data with the spatial coordinates obtained from the object using the SPATA2 package. To add some variability to the spatial distribution (of clones within the same spot), we introduced jitter to the x and y coordinates. We further filtered the data to include only those spots with expression values greater than 3. To create a visually appealing and informative plot, we first plotted the background using the spatial coordinates of the object, with a two-layered approach employing the scattermore package. The first layer consisted of larger black points, and the second layer consisted of smaller white points overlaid on top of the black points to create a subtle border effect. Subsequently, we overlaid the expression data of the target clones on the background, using different colors for each clone and varying the point size based on the expression level. A fixed coordinate system was employed to maintain the spatial integrity of the data. The color palette was generated using the RColorBrewer package, providing a visually distinct set of colors for each clone. Finally, we added spatial index axes using the SPATA2 package, which facilitated the interpretation of the spatial distribution of the clones. This approach allowed us to generate a comprehensive and visually appealing representation of the spatial distribution of the selected clones, revealing their expression patterns and facilitating the identification of potential interactions and relationships within the tissue microenvironment.

### Computation of spatial features of T cell clones
**Total SPTCR UMIs.** To determine the SPTCR parameters we computed the total number of UMI corrected TCR sequences for *n* spots:

$f(x) = \sum_{i=1}^{n} x_i$. For the normalization of TCR-seq data, we first evaluated an RNA-quality and tissue permeabilization index. The challenge in normalization and scaling of array based spatially resolved transcriptomic data is the fact that classical parameters such as total UMIs per spot are relative to the number of cells and the cellular content of these spots (with more or less RNA). Further, heterogeneity in tissue properties lead to significant differences in tissue permiabilisation and finally abundance of RNA molecules that can be captures. To correct for these confounders, we hypothesized that TCR-seq underlies the similar QC and tissue properties therefore we started to quantify the number of UMIs per cells $\frac{\sum_{i=1}^{n} x_i}{c}$ in which the total UMIs (from Illumina sequencing) were divided by the number of cells. This factor was used to normalize the UMI counts per clone and spot. One of the major challenges in normalizing and scaling array-based spatially resolved transcriptomic data lies in the fact that classical parameters, such as the total number of UMIs per spot, are relative to the number of cells and the amount of RNA present in those cells. Moreover, the heterogeneity of tissue properties can result in significant variations in tissue permeabilization, ultimately leading to differences in the abundance of captured RNA molecules. To address these confounding factors, we hypothesized that TCR-seq shares similar quality control (QC) and tissue properties. Therefore, we devised a strategy to quantify the number of UMIs per cell by dividing the total number of UMIs (obtained from Illumina sequencing) by the number of cells $UMI_c \frac{\sum_{i=1}^{n} x_i}{c}$. We then used this factor to normalize the UMI counts per clone and spot

**Number of clones**. The number of clones (per spot) computed number of clones that were found to be expressed at a spot with at least (5 UMI).

**Clonality index**. The getClonaladjacency function calculates the adjacency between TCR clones based on their spatial proximity by employing a series of mathematical procedures. The method can be described as follows: First, Delaunay triangulation is applied to the spatial coordinates $(x_i, y_i)$ of the data in which $i$ represents the number of spots. This process constructs a network of non-overlapping triangles that connects the nearest neighboring points while minimizing the sum of the edge lengths, according to the following equation: $Delaunay(x_i, y_i) \Longrightarrow T = \{T_1, T_2, I, T_n\}$ where $T$ represents the set of triangles formed by the Delaunay triangulation. Next, the adjacency information is extracted from the Delaunay triangulation result, yielding a data frame that stores the relationships between adjacent barcodes $(b)$. For each unique CDR3 region $(CDR3_j)$, the associated barcodes are identified: $CDR3_j = \{b_1, b_2, \ldots, b_m\}$. The resulting data frame is then joined with the adjacency data frame, and the neighboring barcodes for each group are summarized by grouping the data by barcodes. For every barcode in the joined data frame, the number of neighboring barcodes $(NN)$ sharing the same CDR3 sequence is counted: $m = |intersect(b_i, NN_i)|$ If there are no intersect, the $m$ value is set to 0. The number of $|intersect(b_i, NN_i)|$ for each barcode associated with the specific CDR3 region is evaluated by combining: $NN_c = \{CDR3_1 : TCR_1, CDR3_2 : TCRI \ldots, CDR3_k : TCR_k\}$ The getClonaladjacency function returns a data frame containing the clonal adjacency information for all TCR clones based on their spatial proximity. This information can provide valuable insights into the spatial organization of the immune system and its functional implications.

**CDR3-diversity**. The getSpotwiseDiversity function aims to determine the diversity index for each spot by analyzing the TCR information. This analysis requires a series of mathematical procedures that can be broken down into the following steps, while maintaining a coherent narrative: Iterate through each unique barcode $(b_i)$ in the dataset. For each

spot with a specific barcode, perform the following calculations: Within the spot, compare the amino acid sequences of the CDR3 regions $(CDR3_{ij})$ to all other CDR3 regions in the same spot. This comparison is achieved by calculating the string distance, a metric quantifying the dissimilarity between two strings, for each pair of CDR3 sequences: $d_{ijk} = stringdist(CDR3_{ij}, CDR3_{ik})$, for $k = I2, \ldots, n$ Compute the mean string distance for each CDR3 region within the spot by averaging the string distances between the specific CDR3 region and all other CDR3 regions. Calculate the diversity index for the spot by taking the mean of the average string distances for all CDR3 regions within the spot: $DI(x_i) = \frac{1}{n} \sum_{i=1}^{n} d_{ijk}$. By employing these mathematical procedures, the getSpotwiseDiversity function effectively evaluates the diversity of TCR repertoires within each spatial spot.

**Histological classification of spatially resolved transcriptomics**
Histology of each spot was evaluated based on the defined gene expression signature[37] and its histology H&E images. We used the SPATA2 image-annotation tool to segment the images based on histological features. Further, we used the gene expression signatures of the Ivy Gap database[37] to annotate the histological gene expression pattern based on the max likelihood[4].

**Cell type-specific gene expression and metabolism**
We computed spatially resolved cell type-specific gene expression as reported recently in the RCTD pipeline. The getCelltypeSpecificGeneExpression function processes single or multiple cell types and enhances cell type-specific gene expression patterns in a given object. The procedure involves several mathematical steps, which are explained below. First, the Iancer (e) factor for each gene is computed as follows: $e_i = avg_{\log 2FCi} + 1$. A submatrix $(M_o)$ is extracted from the original expression matrix (M) containing only the significant genes: $M^0 = M[g^1, g^2, \ldots, g_n, :]$, where $g^1, g^2, \ldots, g_n$ are the indices of the significant genes. Next, the product of the transpose of the submatrix and the diagonal matrix of enhancer factors is calculated: $\Delta M = |M^{0T}| \times diag(e^z)$ The expression values for the significant genes are updated in the matrix: $M[g^1, g^2, \ldots, g_n, :] = M^0 + \Delta M^T$ For the single cell type case, the feature-specific scaling factor $(f)$ is computed using the exponential function: $f = e^{x^1, x^2, \ldots, x_n}$, where $x^1, x^2, \ldots, x_n$ are the expression values for the cell type. The scaling factor $(f)$ is then updated: $f' = f + (f-1) \times e$. For the multiple cell types, the scaling factor $(f)$ is calculated as the average of the exponential values for each cell type. Finally, the expression matrix is updated with the scaling factor: $M = M + |M|^T \times f'$ The resulting expression matrix (M) highlights the cell type-specific gene expression patterns and can be used for further analysis.

**Node-centric expression modeling for cell–cell communication**
In this study, we employed Node Centric Expression Modeling (NCEM) for assessing cell-cell communication within the tumor microenvironment. We crafted layered AnnData objects encompassing lymphoid and myeloid cell types, malignant cells, and stromal cells, all aligned to the initial barcodes from cell-specific expression determined through cell2location. To evaluate T cell function within both 'productive' and 'exhausted' regions of the tumor, we classified spots based on the expression of clonal and exhaustion markers in CD8 T cells. Clonal markers included *IL2, IFNG, PRF1, GZMB, GZMK, GZMA, CD69, CD25, and CD38*, while exhaustion markers encompassed *HAVCR2, PDCD1, CTLA4, CXCL8, LAG3, EOMES, TOX, TIGIT, CD244, and NR4A1*. Along with this, TCR presence was factored in. We formulated a normalized composite score from these variables, ultimately differentiating "productive" and 'exhausted' spots. Subsequently, for each sample, we created AnnData objects for exhausted and reactive niches that included the top 2000 highly variable genes of the estimated gene expression. With the Visium-specific InterpreterDeconvolution model, NCEM leveraged these AnnData objects to determine niche-specific gene expression

from cell composition and cell-specific gene expression. To visualize the results, we developed circular graph plots. We combined the 'estimated magnitude' of each cell type's interaction, added the min–max scaled matrices for every niche and cell type to a total, and performed another scaling to achieve a cumulative weight of each cell type's interaction for expression. After pruning low-impact edges (de_genes <0.3), we plotted the network graph, with the cell population-colored edges adjusted in intensity to match the weight of the interaction. The focus was placed on interactions of significant relevance.

## Hierarchical cell type composition model (HCCM)

The createCellGraph function is designed to construct a cell graph based on spatial coordinates and a set of features using Delaunay triangulation. Delaunay triangulation is a method that creates a set of non-overlapping triangles, $T = \{t_1, t_2, \ldots, t_m\}$, from a set of points $P = \{p_1, p_2, \ldots, p_n\}$ in the plane. This method ensures that for each triangle $t_j \in T$, there exists a circumcircle passing through its vertices $p_{j1}$, $p_{j2}$, $p_{j3}$, and no other point from the set $P$ lies inside that circle. The Delaunay triangulation maximizes the minimum angle of all triangles in the set, resulting in a well-distributed configuration. By applying the Delaunay triangulation method to the spatial coordinates (x, y) of the cells, the createCellGraph function generates a set of edges that represent the spatial relationships between cells. The resulting graph provides a structured representation of the relationships between cells, considering both their spatial proximity and the selected features. To create the cell graph, the function first extracts the spatial coordinates of the cells corresponding to the given features. Then, it computes the Delaunay triangulation on these coordinates, generating a set of edges connecting the nodes (cells) based on their spatial proximity. The function associates each edge with a pair of barcodes to establish a correspondence between the cells. Finally, it merges the edge information with the spatial coordinates, resulting in a data frame containing the source and destination nodes of each edge based on the Delaunay triangulation. The output of the createCellGraph function is a data frame representing the cell graph with edges derived from the Delaunay triangulation of the spatial coordinates. This graph allows for the exploration and analysis of the relationships between cells in a spatial context, considering the selected features. We transformed the graph into an igraph[38] object and computed the relationships between data points in a hierarchical manner, where data points are organized into nested levels. These types of graphs are useful for visualizing relationships and understanding the structure of data sets that have inherent hierarchies such as cellular interactions and compositions. We decided to visualize hierarchical graphs through circle packing.

## Receptor ligand interaction

We employed an advanced approach to analyze receptor-ligand interactions between cells using the CellChat package. We extracted the human CellChat database (CellChatDB.human) to obtain information on ligand-receptor interactions. To explore the interactions between B cells and NK cells, we focused on the spatial gene expression patterns of these cell types. We utilized the getCelltypeSpecificGeneExpression function to generate cell-type-specific gene expression matrices for B cells, NK cells, and CD4+ interferon (IFN)-expressing cells with enhanced expression patterns, leveraging the SPATA2 package. Next, we filtered the ligand-receptor interaction data to include only those interactions involving genes present in the gene expression matrices of B cells and NK cells. This step allowed us to focus on the most relevant interactions for our analysis. For each ligand-receptor interaction, we calculated the Kullback-Leibler divergence (KLD) between the expression patterns of the ligand and receptor genes. This step facilitated the identification of potential spatial relationships between these genes and provided insight into the strength and directionality of their interactions. Lastly, we visualized the top 100 interactions with the lowest KLD scores, showcasing

the most significant spatial relationships between B cells and NK cells in our dataset. The resulting plot provided a comprehensive view of the receptor-ligand interactions and facilitated a deeper understanding of the complex interplay between these cell types within the tissue microenvironment. By combining spatial gene expression analysis with receptor-ligand interaction data, our approach allowed for a thorough exploration of the molecular crosstalk between B cells and NK cells. This information can further our understanding of the underlying mechanisms driving cellular communication and immune cell behavior in the tissue microenvironment.

## Cell type deconvolution

**Robust cell type decomposition.** Cell type deconvolution was performed by spacexr (Spatial-eXpression-R): Robust Cell Type Decomposition (RCTD)[39] which is implemented in the SPATAwrappers (https://github.com/heilandd/-SPATAwrappers) package (runRCTD). For the single cell reference, we used the GBMap atlas consist of more than 1 M cells[40]. The following parameters were chosen: -cell_type_var = "annotation_level_4".

**CytoSpace.** For the single-cell deconvolution utilizing Cytospace and SPATA objects, we have developed a streamlined pipeline accessible at https://github.com/heilandd/SPTCR-seq. T"e R script, "CytoSpace_"rom_SPATA.R," delineates a comprehensive protocol for exporting the necessary file format compatible with the Cytospace pipeline, as well as a bash script for automated processing of multiple SPATA2 objects. Cytospace analysis is performed in a bash environment. Upon completion of the analysis, we supply a script for importing the results. The Cytospace output is integrated into existing SPATA2 objects through the CytoSpace2SPATA function, available at https://github.com/heilandd/-SPTCR-seq. For the deconvolution process, we employed the GBMap atlas[40].

**Cell2Location.** In this study, we integrated the recently published GBMap single-cell dataset of glioblastoma with our Visium spatial transcriptomics data. We downsampled the single cell dataset to 100k to enable computations. First, we estimated signatures from cells from the single-cell dataset using the cell2location Negative Binomial regression model (cell2location.models.RegressionModel()) which generates the inf_aver_sc.csv table used for spatial deconvolution. Next, we identified shared genes between the signature genes and the spatial dataset, initializing the cell2location model (cell2location.models.Cell2location). We trained the model with recommended hyperparameters, using early stopping based on ELBO loss. After training, we estimated and exported the posterior distribution of cell abundance for further analysis using the mod.export_posterior() function. We computed the expected expression per cell type using the mod.module.model.compute_expected_per_cell_type() function and exported cell-specific expression. Finally, we created an NCEM object containing the cell expression of relevant cell populations, proportions, node types, and spatial information for downstream analysis.

## Annotation of clones to states or spatial transcriptional programs

For deconvolution of T cell subtypes, we utilized our T cell atlas with annotated T cell states from glioblastoma IDH wild-type patients. The annotation contains groups for CD4 (6 cell states) and CD8 (6 cell states) spanning effector to T cell dysfunction[7]. To annotate the T cell state to each clone derived from the SPTCR-seq, we estimated the likelihood of each T cell states by RCTD ($s_{1,...,n}$) for each clone ($c_{1,...,n}$) resulting in a $s \times c$ matrix in which $s$ is calculated by

$$f_{(s,n)} = \begin{cases} s & n = 1 \\ \sum_{i=1}^{n} s_i & n > 1 \end{cases}$$ with $n$ representing the number of spots in

which the clone $c$ can be detected. This matrix was used for

dimensional reduction with Uniform Manifold Approximation and Projection[41]. A similar approach was chosen to define the transcriptional niches of each clone. A matrix of $c \times T$ was computed in which $c$ was the clone and $T$ the averaged scores of each spatial distinct transcriptional program[4]. The matrix ($c \times T$) was visualized by frequency of clones per transcriptional program by a mosaic plot using the *geom_mosaic* function.

## Spatially weighted regression analysis

For spatially weighted correlation analysis we used the function *SPATAwrappers::runSpatialRegression* with the following parameters: In case of multi parameter comparisons ($n > 10$) we used the model: canonical-correlation analysis (CCA), for lower number of parameters we applied the Spatial Durbin linear (SLX, spatially lagged X)[42] or the Spatial simultaneous autoregressive lag model[43]. The parameters smooth and normalize were set as false. The spatial regression analysis will provide a matrix $n \times n$ in which $n$ are the selected variables with corresponding $x$ and $y$ positions and estimated neighbors.

## Code and language optimization

In this study, we employed multiple large language models (LLMs), including ChatGPT, to enhance computational efficiency, streamline code explanation and documentation, and refine the wording in the results section. We specifically integrated ChatGPT into R software to optimize and restructure the R code. Additionally, we utilized ChatGPT and other LLMs to accurately translate computational processes into coherent and precise written text within the methods section.

## Statistics and Reproducibility

In this study, histological annotations and local clonal expansion analyses were conducted on a selection of nine primary and recurrent glioblastoma samples, establishing the basis for robust findings. Evidence supporting these analyses can be seen in various figures throughout the study. Specifically, Figs. 1b, 3c, and 4d present surface plots that demonstrate the interplay between histological patterns and T cell infiltration. Additionally, Fig. 5f, g depict local T cell clonal expansion patterns, elucidating our methodology further. Figures 5d and 2f offer relative and absolute quantitative analyses respectively; the former displays the distribution of histology presence across all samples, while the latter provides a count of TCR_UMIs per sample across all histology subtypes. It should be noted that the sample size was not determined using a statistical method, and all data collected were included in the analyses. The investigators were not blinded to allocation during experiments or outcome assessment, and the experiments were not randomized.

## Reporting summary

Further information on research design is available in the Nature Portfolio Reporting Summary linked to this article.

## Data availability

The SPTCR-seq fastqs and processed files are available under the GEO accession code "GSE238071". The spatial transcriptomics data used in this study has been deposited on DataDryad and is accessible to the public (https://doi.org/10.5061/dryad.h70rxwdmj). The spatial metabolomics data can be found on the OSF platform using this link: https://osf.io/8qbdz/?view_only=5287d7f6263e4ba680ca8c396aeefeee. Further processed files and detailed steps of our analysis have also been made available on OSF: https://osf.io/65y3t/?view_only=6571f0c374ce4bf294b9cbd10ade62cf. The source files for Figs. 1 and 2 can be found on the OSF platform using this link: https://osf.io/8qbdz/?view_only=5287d7f6263e4ba680ca8c396aeefeee. Source data for Figs. 2f,g, 4b,c, 5b,c, 5f, 7b,c, 7f,g, and 8b,c are provided as a Source Data file. Source data are provided with this paper.

## Code availability

The version of our SPTCR pipeline used in this study can be accessed via https://doi.org/10.5281/zenodo.8161782. A maintained version of this tool is available at https://github.com/theMILOlab/SPTCR-Seq-Pipeline. In addition, this study resulted in a software tool named SPATAImmune, which can be accessed on GitHub: https://github.com/theMILOlab/SPATAImmune. The scripts used for data analysis are accessible on GitHub at https://github.com/heilandd/SPTCR_seq_code (https://doi.org/10.5281/zenodo.8161784). With the provided code and scripts, all figures presented in the study can be reproduced.

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

## Acknowledgements

This project was funded by the German Cancer Consortium (DKTK) (D.H.H.), Else Kröner-Fresenius Foundation (D.H.H.). The work is part of the MEPHISTO project (D.H.H.), funded by BMBF (iGerman Ministry of Education and Research) (project number: 031L0260B).

We would like to acknowledge the use of BioRender.com (licenses: "Student Plan" by D.H.H.) for the creation of illustrations present in this publication. Specifically, parts of Figs. 1a, 2a, 3a, 4a, 6a, 7a, and 8a were generated with BioRender and created by D.H.H.

## Author contributions

D.H.H. conceived the idea and designed the experiments. D.H.H. and J.K.B. composed the paper, J.K.B. conducted the library preparation, sequencing as well as benchmarking of the method. J.K.B. conducted the development of the SPTCR software pipeline and Niche-specific expression analysis. J.K. and D.H.H. conducted analysis and software design of SPATAImmune, D.H.H., J.K.B. conducted the data analysis and interpretation. O.S., C.L.C., R.S., J.B., J.Z. V.M.R., P.W., K.J. contributed to the interpretation of the results, D.H.H. supervised the project.

## Funding

## Competing interests

The authors declare no competing interests.
