## [Peer Review File · Nature Communications]

Reviewers' comments:

Reviewer #1 (Remarks to the Author):

The authors introduce SPTCR-seq, an integration of commercially available spatial transcriptome (ST) sequencing protocol (10x visium as the demo) and RAGE-seq protocol, to provide spatially resolved TCR and transcriptome information. They also contribute a series of bioinformatic pipelines including error correction and TCR reconstruction to make the protocol easy to use for the community. SPTCR-seq was applied to 9 glioblastoma tumor samples as a demo case to demonstrate the utility of the methodology. I think the protocol proposed is workable as both RAGE-seq and ST protocols are widely used, but it is still better to add a table to compare systematically SPTCR and other published spatial TCR methods in important aspects, including sequencing depth (bases) as well as costs. The biggest problem seems to lie in the second part of the study, which I am a bit disappointed with the GBM data analysis part. I think the authors didn't show us the power of spatially resolved single cell TCR and transcriptome technology or SPTCR-seq, rather, most of the analysis provided could be derived by single-cell ST without TCR sequences. There are some interesting results in this part such as TCR clonality and diversity correlates with neighboring cell subsets, but the entire part is overall descriptive and without explanations or discussions.

1. L96, there are not too much meaning to compare with MiXCR or TRUST4, as they are not designed to handle single cell and long-reads (error enriched) data;
2. The authors should carefully check out all the figures and legends throughout the manuscript, as there are a lot of errors. Citations of supplementary figure 4 of L121-128 are full of errors; no citation of figure 2d etc. All these make the readers difficult to follow.
3. In figure 2 and 3, authors may try to clarify to the readers what additional information can be provided by spatially resolved TCR sequences. High-resolution single cell ST (10X visium hasn't achieved single-cell resolution) or usual single-cell RNA+TCR-seq may provide similar information in Figure 2 and figure 3?

Reviewer #3 (Remarks to the Author):

It was a pleasure for me to read the manuscript, just because it is amazing how the field is developing. We always dreamed about tracking the T cells clones in tissues, now it is becoming the reality.

However, the major advance here is the Visium solution from 10x Genomics, per se. When the barcoded cDNA library is obtained and amplified, it is a matter of simple technique how to get the TCR libraries from it. Hudson et al (STAR Protocols, 2022) and Liu et al (Immunity, 2022) did it with a set of multiplex V segment specific oligos, which is the simplest solution. Considering that all the bottlenecks (cDNA synthesis and cDNA entering the 1st PCR amplification) are already left behind (which is all within a standard Visium protocol), and considering that UMI and localization barcode clustering allows to eliminate possible minor (at this stage) biases of the multiplex PCR, this should work absolutely fine.

The TCR capture approach used by authors of the current work looks a bit more laborious, but also possible. However, the reviewer does not find any novelty in this (minor) modification, and does not see any reason, why such approach could be more sensitive compared to the simplest multiplex PCR in this case. Their approach then leads them to limitations of the nanopore platform with low base call accuracy of long-read sequencing, and the authors then fight with these issues computationally, but all this does not increase the sensitivity, for sure.

The authors claim “an order of magnitude higher sensitivity than previously described protocols”. However, first, back-to-back comparison was not performed. And second, technically, there are no explanations why the “target enrichment probe hybridization” approach for the ready pre-amplified cDNA library (which already contains starting cDNA molecules in a million of copies) should be more sensitive than the simple multiplex PCR.

Also, surprisingly (disappointingly) authors do not show the images of clonal T cell distribution (as, for example, Figure 3 of Liu et al, Immunity, 2022). Here we can only see the Figure 1d, without clonal annotations. Such simple clear figures are much more persuading than further computational derivatives, to reviewer opinion.

Altogether, while reviewer is happy to see the field developing, and highly appreciates the efforts of the authors, he has to admit that he does not see a sufficient level of novelty and technical advantage in the proposed method.

A couple of minor comments:

1) In the abstract: “The SPTCR computational pipeline including spatial mapping, error and UMI correction, approaches the yield and coverage of other single cell TCR technologies (>1,000 reads per chain).”

Not clear here what do authors mean by “reads per chain”. Sequencing reads are just reads, you can get as many as you pay. True unique cDNA molecules captured is a measure of sensitivity. Probably reviewer misunderstood this, better reformulate.

2) "utilized 45 pooled TRBV forward primers or RNase H-dependent PCR (rhTCR-seq) from the TCR a and b regions to enrich for full length TCR sequences."

Not “full length” here, since multiplex TRBV anneals on the FR3. So probably better: “to enrich for CDR3-containing TCR chain fragments”.

Reviewer #1 (Remarks to the Author):

The authors introduce SPTCR-seq, an integration of commercially available spatial transcriptome (ST) sequencing protocol (10x visium as the demo) and RAGE-seq protocol, to provide spatially resolved TCR and transcriptome information. They also contribute a serial of bioinformatic pipelines including error correction and TCR reconstruction to make the protocol easy to use for the community. SPTCR-seq was applied to 9 glioblastoma tumor samples as a demo case to demonstrate the utility of the methodology. I think the protocol proposed is workable as both RAGE-seq and ST protocols are widely used, but it is still better to add a table to compare systematically SPTCR and other published spatial TCR methods in important aspects, including sequencing depth(bases) as well as costs.

We thank the reviewer for its overall positive comments and have performed an experimental comparison of all published spatially resolved TCR sequencing protocols. We also added a brief overview of the costs and labor which is required for each method:

The following paragraphs are added to the manuscript:

Comparative Analysis of SPTCR-seq and PCR-based Spatial TCR-seq Methods

In the next phase of our study, we aimed to compare SPTCR-seq with published protocols to evaluate their respective advantages and disadvantages. Both PCR-based protocols (Hudson and Liu) are

considerably more cost-effective and less labor-intensive than SPTCR-seq: hands-on time for Hudson and Liu protocols is 4-6 hours, while for SPTCR-seq, it is 6-8 hours plus an overnight incubation. The cost per sample (excluding sequencing) is approximately 5 Euros for the Hudson protocol, 12 Euros for the Liu protocol (2.4 fold-change), and 32 Euros for SPTCR-seq (6.4 fold-change). It should be noted that the estimated hands-on time and cost per sample may vary if samples are multiplexed together. The primary cost difference can be attributed to the library preparation workflow and the degree of sample multiplexing, as Nanopore technology tends to be significantly more expensive than Illumina when samples are not multiplexed. To compare the protocols, we used three Visium libraries and performed all three protocols as recently described. The sequencing depth from Hudson and Liu was approximately 300 million reads/sample, and the SPTCR-seq yield was around 26-30 million reads. All protocols were processed by the individual pipelines, as described in **Figure 3a**. We began our comparison by examining the diversity of distinct TCRs detected by each method. To this end, we evaluated the number of unique CDR3s per sample by applying a stringent criterion that classified CDR3 regions as unique if they exhibited a Levenshtein distance of at least two. Due to the lack of TCR α chains in the Hudson protocol, we focused our comparison on the TCR β chain. The Liu protocol identified a mean of 13.4 unique TRBs (min=4, max=31, n=5) per sample, whereas the Hudson protocol yielded a mean TRB count of 26, ranging from 16 to 36 TCRs per sample (n=3). In comparison, SPTCR-seq detected a mean of 208 unique TRBs (sd:135) per sample (n=5). For combined TRA & TRB counts, the Liu protocol achieved a mean of 16.2 (min 6, max 33, n=5) in matched samples, compared to SPTCR's 208.6 unique TRA/B CDR3 regions (min 24, max 343). Our data showed that SPTCR-seq maintained a more diverse immune receptor repertoire, with a mean of 87.25 UMIs per TRB TCR, as opposed to 1.2 UMIs per TCR across all samples for the Hudson protocol (n=3) and Liu protocol (n=7). To investigate the reasons behind the lower number of detected TCRs in PCR-based methods compared to SPTCR-seq, we visualized the mapped segments reported by MixCR and SPTCR-seq in a waterfall plot, **Figure 3b**. The TCR reconstruction algorithm successfully mapped 93% of reads to a TCR locus for the Hudson protocol (n=3) and 56% for the Liu protocol (n=7), while IgBlast mapped SPTCR-seq reads to a TCR locus in 98.4%. Subsequently, we visualized the spatial distribution of TCRs and observed a consistent spatial abundance pattern of unique TRB UMIs across all methods, **Figure 3c-d**. The main limitation of PCR-based methods is their inability to accurately annotate J regions (approximately 92% failure rate for the Hudson protocol and 55% for the Liu protocol), which results in a significant loss of reads necessary for complete TCR reconstruction. In contrast, SPTCR-seq successfully annotates V and J regions 87.56% of the time. Although the final anchoring of CDR regions still presents challenges, SPTCR-seq attains a considerably higher rate of full TCR reconstruction compared to other methods. We primarily attribute this significant difference to the shorter read length of the 300 Cycles v3 Illumina chemistry. Although Illumina's high throughput excels at capturing gene expression at high coverage, the short read length hampers its ability to span the structurally important regions of the TCR. In conclusion, our comparison of SPTCR-seq with the Hudson and Liu protocols demonstrated that SPTCR-seq outperforms these PCR-based methods regarding TCR diversity and annotation accuracy. Despite the higher cost and increased hands-on time associated with SPTCR-seq, its sensitivity to detect a more diverse immune receptor repertoire and annotate full TCR transcripts makes it a valuable tool for TCR-seq analysis. Additionally, the robustness and scalability of the SPTCR-

seq method make it a promising approach for future studies aiming to examine the spatial distribution and evolution of T cell receptor sequences in various research contexts.

Figure 3: a) Illustration of the comparison of three methods. b) Waterfall plots of the three methods indicate the loss of reads during the TCR annotation and reconstruction. c-d) Surface plot indicates the histology (c) and the spatial distribution of annotated TCRs across methods.

The biggest problem seems to lie in the second part of the study, which I am a bit disappointed with the GBM data analysis part. I think the authors didn't show us the power of spatially resolved single cell TCR and transcriptome technology or SPTCR-seq, rather, most of the analysis provided could be derived by single-cell ST without TCR sequences. There are some interesting results in this part such as TCR clonality and diversity correlates with neighboring cell subsets, but the entire part is overall descriptive and without explanations or discussions.

We appreciate the reviewer's critique and sincerely apologize if the initial focus of our manuscript did not meet expectations. The manuscript was originally designed as a short report for another journal and was later transferred to 'Nature Communications'. Due to the space constraints associated with the initial plan, our emphasis was primarily on the methodology rather than the in-depth analysis of Glioblastoma Multiforme (GBM) data. We acknowledge the reviewer's concerns about the second part of our study, particularly regarding the perceived lack of unique insights offered by our method. We agree that there are significant opportunities to delve deeper into the data analysis, revealing more comprehensive insights into the potential power of SPTCR-seq. In response to this feedback, we have now expended substantial effort towards a more in-depth characterization of spatial TCR. We fully renewed our analysis

part implemented recent state-of-the-art models and focused on the unique power of spatially resolved data. Further we integrated metabolomic data (MALDI) to showcase the power of our method.

1. L96, there are not too much meaning to compare with MiXCR or TRUST4, as they are not designed to handle single cell and long-reads (error enriched) data;

We agree with the reviewer and removed this part of the manuscript.

2. The authors should carefully check out all the figures and legends throughout the manuscript, as there are a lot of errors. Citations of supplementary figure 4 of L121-128 are full of errors; no citation of figure 2d etc. All these make the readers difficult to follow.

The worked on improve the concerns in our revised version of the manuscript.

3. In figure 2 and 3, authors may try to clarify to the readers what additional information can be provided by spatially resolved TCR sequences. High-resolution single cell ST (10X visium hasn't achieved single-cell resolution) or usual single-cell RNA+TCR-seq may provide similar information in Figure 2 acd and figure 3?

We appreciate this concern and focus in the revised version on the power of spatially resolved technologies.

Reviewer #3 (Remarks to the Author):

It was a pleasure for me to read the manuscript, just because it is amazing how the filed is developing. We always dreamed about tracking the T cells clones in tissues, now it is becoming the reality. However, the major advance here is the Visium solution from 10x Genomics, per se. When the barcoded cDNA library is obtained and amplified, it is a matter of simple technique how to get the TCR libraries from it. Hudson et al (STAR Protocols, 2022) and Liu et al (Immunity, 2022) did it with a set of multiplex V segment specific oligos, which is the simplest solution. Considering that all the bottlenecks (cDNA synthesis and cDNA entering the 1st PCR amplification) are already left behind (which is all within a standard Visium protocol), and considering that UMI and localization barcode clustering allows to eliminate possible minor (at this stage) biases of the multiplex PCR, this should work absolutely fine. The TCR capture approach used by authors of the current work looks a bit more laborious, but also possible. However, the reviewer does not find any novelty in this (minor) modification, and does not see any reason, why such approach could be more sensitive compared to the simplest multiplex PCR in this case. Their approach then leads them to limitations of the nanopore platform with low base call accuracy of long-read sequencing, and the authors then fight with these issues computationally, but all this does not increase the sensitivity, for sure. The authors claim "an order of magnitude higher sensitivity than previously described protocols". However, first, back-to-back comparison was not performed. And second, technically, there are no explanations why the "target enrichment probe hybridization" approach for the ready pre-amplified cDNA library (which already contains starting cDNA molecules in a million of copies) should be more sensitive than the simple multiplex PCR.

We appreciate the reviewer's insightful comment and have taken the suggestion to conduct a comprehensive comparison of all protocols. Our findings revealed significant differences across all three methods. A primary distinction we observed between PCR-based methods and target enrichment via hybridization involves the number of PCR cycles required, which recent studies have identified as a potential source of errors (as discussed in the results part). Our analysis indicated that the frequency of fully annotated TCRs was significantly less efficient when utilizing PCR and Illumina protocols as compared to SPTCRseq. We also found that the recent improvements in Nanopore chemistry and the associated reduction in sequencing errors (which we have also evaluated) greatly enhance postprocessing and TCR annotations. However, it should be noted that hybridization and nanopore sequencing have the disadvantage of higher per-sample costs and labor intensity, details of which we have elaborated upon in our manuscript. The choice of protocol may also be influenced by the specific type of cancer or tissue under investigation. For instance, Glioblastomas (GBMs) are characterized as 'cold' tumors, necessitating a greater sensitivity of target enrichment. This requirement may not be as stringent for immune-rich cancers or tissues that do not demand enhanced amplification. We trust that our comparative analysis will aid in evaluating the various methods, illuminating their respective strengths and weaknesses. We have added a new section to our manuscript addressing this topic:

Comparative Analysis of SPTCR-seq and PCR-based Spatial TCR-seq Methods

In the next phase of our study, we aimed to compare SPTCR-seq with published protocols to evaluate their respective advantages and disadvantages. Both PCR-based protocols (Hudson and Liu) are considerably more cost-effective and less labor-intensive than SPTCR-seq: hands-on time for Hudson and Liu protocols is 4-6 hours, while for SPTCR-seq, it is 6-8 hours plus an overnight incubation. The cost per sample (excluding sequencing) is approximately 5 Euros for the Hudson protocol, 12 Euros for the Liu protocol (2.4 fold-change), and 32 Euros for SPTCR-seq (6.4 fold-change). It should be noted that the estimated hands-on time and cost per sample may vary if samples are multiplexed together. The primary cost difference can be attributed to the library preparation workflow and the degree of sample multiplexing, as Nanopore technology tends to be significantly more expensive than Illumina when samples are not multiplexed. To compare the protocols, we used three Visium libraries and performed all three protocols as recently described. The sequencing depth from Hudson and Liu was approximately 300 million reads/sample, and the SPTCR-seq yield was around 26-30 million reads. All protocols were processed by the individual pipelines, as described in **Figure 3a**. We began our comparison by examining the diversity of distinct TCRs detected by each method. To this end, we evaluated the number of unique CDR3s per sample by applying a stringent criterion that classified CDR3 regions as unique if they exhibited a Levenshtein distance of at least two. Due to the lack of TCR α chains in the Hudson protocol, we focused our comparison on the TCR β chain. The Liu protocol identified a mean of 13.4 unique TRBs (min=4, max=31, n=5) per sample, whereas the Hudson protocol yielded a mean TRB count of 26, ranging from 16 to 36 TCRs per sample (n=3). In comparison, SPTCR-seq detected a mean of 208 unique TRBs (sd:135) per sample (n=5). For combined TRA & TRB counts, the Liu protocol achieved a mean of 16.2 (min 6, max 33, n=5) in matched samples, compared to SPTCR's 208.6 unique TRA/B CDR3 regions (min 24, max 343). Our data showed that SPTCR-seq maintained a more diverse immune receptor repertoire, with a mean of 87.25 UMIs per TRB TCR, as

opposed to 1.2 UMIs per TCR across all samples for the Hudson protocol (n=3) and Liu protocol (n=7). To investigate the reasons behind the lower number of detected TCRs in PCR-based methods compared to SPTCR-seq, we visualized the mapped segments reported by MixCR and SPTCR-seq in a waterfall plot, **Figure 3b**. The TCR reconstruction algorithm successfully mapped 93% of reads to a TCR locus for the Hudson protocol (n=3) and 56% for the Liu protocol (n=7), while IgBlast mapped SPTCR-seq reads to a TCR locus in 98.4%. Subsequently, we visualized the spatial distribution of TCRs and observed a consistent spatial abundance pattern of unique TRB UMIs across all methods, **Figure 3c-d**. The main limitation of PCR-based methods is their inability to accurately annotate J regions (approximately 92% failure rate for the Hudson protocol and 55% for the Liu protocol), which results in a significant loss of reads necessary for complete TCR reconstruction. In contrast, SPTCR-seq successfully annotates V and J regions 87.56% of the time. Although the final anchoring of CDR regions still presents challenges, SPTCR-seq attains a considerably higher rate of full TCR reconstruction compared to other methods. We primarily attribute this significant difference to the shorter read length of the 300 Cycles v3 Illumina chemistry. Although Illumina's high throughput excels at capturing gene expression at high coverage, the short read length hampers its ability to span the structurally important regions of the TCR. In conclusion, our comparison of SPTCR-seq with the Hudson and Liu protocols demonstrated that SPTCR-seq outperforms these PCR-based methods regarding TCR diversity and annotation accuracy. Despite the higher cost and increased hands-on time associated with SPTCR-seq, its sensitivity to detect a more diverse immune receptor repertoire and annotate full TCR transcripts makes it a valuable tool for TCR-seq analysis. Additionally, the robustness and scalability of the SPTCR-seq method make it a promising approach for future studies aiming to examine the spatial distribution and evolution of T cell receptor sequences in various research contexts.

Figure 3: a) Illustration of the comparison of three methods. b) Waterfall plots of the three methods indicate the loss of reads during the TCR annotation and reconstruction. c-d) Surface plot indicates the histology (c) and the spatial distribution of annotated TCRs across methods.

Also, surprisingly (disappointingly) authors do not show the images of clonal T cell distribution (as, for example, Figure 3 of Liu et al, Immunity, 2022). Here we can only see the Figure 1d, without clonal annotations. Such simple clear figures are much more persuading than further computational derivatives, to reviewer opinion.

We have extensively worked on improve the application of our method to explore the spatial T cell heterogeneity and distribution in GBM. We also added requested illustrations of spatial clonal expansion. The following plots are added to the manuscript:

Local clonal diversity is associated with tumor-associated T cell dysfunction/exhaustion.

Utilizing our established parameters, we proceeded to categorize each spot as "local T cell expansion", "local T cell diversity" and "no detected T cells" or spots that could not be unambiguously classified, **Figure 5a**. In the analysis of all samples, clonal T cell expansion was observed in only 6 out of 9 samples, in a minority of spots ranging from 2 to 68 spots, while T cell diversity was detected in every sample, **Figure 5b**. Furthermore, no distinct histological regions could be associated with local clonal expansion or clonal diversity, **Figure 5c**. We postulated that the local expansion of clones is strongly correlated with regions with potential antigens or favorable microenvironmental conditions. Since unique clonal expansion was only rarely observed, indicating that T cells in glioblastoma do not display a single dominating T cell clone; instead, they exhibit T cell expansion independent of a specifically presented antigen. This hypothesis is substantiated by the spatial annotation of clones exhibiting a high Clonality-Index, which revealed a significant overlap of local T cell expansion within the same tumor regions, **Figure 5e-f**. Based on these findings, we aimed to investigate to what extent these clones are functionally active to support anti-tumor immunity or dysfunctional/exhausted. Using supervised spatial clustering based on predefined T cell cytotoxic and dysfunctional/exhausted signatures, we identified that large parts of the "local T cell diversity" pattern showed enriched expression of exhausted genes. Some parts of the "local T cell diversity" along with the "local T cell expansion" pattern showed upregulation of classical cytotoxic markers suggesting that only a minority of the detected clones contribute to anti-tumor immune response.

Figure 5: a) Representative example of the sample UKF259. Colors indicate the TCR label of the spot. b) Stacked barplot of the distribution of T cell response classes (indicated by colors) across samples. c) Barplot of the number of spots classified as clonal. d) Stacked barplot of the distribution of histological groups across T cell response classes. e-g) Surface plots with histology (left) and spatial mapping of single clones.

Altogether, while reviewer is happy to see the field developing, and highly appreciates the efforts of the authors, he has to admit that he does not see a sufficient level of novelty and technical advantage in the proposed method.

We hope to address this concern in the revised version of our manuscript.

A couple of minor comments:

1) In the abstract: “The SPTCR computational pipeline including spatial mapping, error and UMI correction, approaches the yield and coverage of other single cell TCR technologies (>1,000 reads per chain).” Not clear here what do authors mean by “reads per chain”. Sequencing reads are just reads, you can get as many as you pay. True unique cDNA molecules captured is a measure of sensitivity. Probably reviewer misunderstood this, better reformulate.

We agree with the reviewer on the unclear statement in the abstract and have changed the abstract as follows:

Spatial resolution of the T cell repertoire is essential for deciphering cancer-associated immune dysfunction. Existing array-based spatially resolved transcriptomic technologies are unable to directly annotate T cell receptors (TCR). We present spatially resolved T cell receptor sequencing (SPTCR-

seq), which integrates optimized target enrichment and long-read sequencing for highly sensitive TCR sequencing, attaining an on-target rate of ~85%. The SPTCR computational pipeline, encompassing spatial mapping, error correction, and UMI refinement, achieves yield and coverage comparable to alternative single-cell TCR technologies. Our extensive comparison of PCR-based and SPTCR-seq methods underscores SPTCR-seq's superior ability to reconstruct the entire TCR architecture, including V, D, J regions and the pivotal complementarity-determining region 3 (CDR3). Employing SPTCR-seq, we assessed local T cell diversity and clonal expansion and transcriptional evolution across spatially discrete niches. Further investigation into the microenvironmental impact on exhausted and cytotoxic T cells disclosed the critical involvement of NK and B cells in spatial T cell adaptation. The synergy of spatially resolved omics and SPTCR-seq proves advantageous across diverse experimental scenarios, facilitating the exploration of T cell dysfunction in cancers and other diseases.

2) "utilized 45 pooled TRBV forward primers or RNase H-dependent PCR (rhTCR-seq) from the TCR a and b regions to enrich for full length TCR sequences." Not "full length" here, since multiplex TRBV anneals on the FR3. So probably better: "m".

We fully concur. Due to the restructuring of the manuscript, this particular sentence will not be incorporated further.

REVIEWERS' COMMENTS

Reviewer #1 (Remarks to the Author):

I think the revised manuscript has substantially addressed most of my concerns and significantly improved. The authors did a comparison of SPTCR-seq with other spatial-TCR methods, and reconstructed more TCR clonotypes per sample by their method, though with higher cost and longer hands-on time. A head-to-head comparison of various methods on same samples is definitely welcome, although I have reservation on the design and conclusions made in this manuscript. The authors explained that the differences of performance mostly come from the long read-length of SPTCR-seq and shorter read-length of multiplex-PCR based methods, however, the PCR based method can also be adapted to long read length by nanopore sequencing. Therefore, it is an unfair comparison and should be noted in the manuscript.

The authors added substantial analysis of adopting spatially-resolved TCR data and other multi-omics data, such as transcriptome and metabolome data, to link the T cell expansion and diversity with patterns of cellular interactions, as well as T cell dysfunction or cytotoxicity. I think these analyses added more value in the revised manuscript.

Some details are listed below.

1. L283-286, there are no supporting data/figures for these results.
2. L350-351, citations should be added here.
3. L373-374, description of “additional layer of metabolomics” should be added, or a proper citation with the detailed description can be added.
4. L415-417, these sentences should be rephrased, as noted by the argument I mentioned above, also, PCR cycle reduction does not lead to higher sensitivity?

Reviewer #3 (Remarks to the Author):

I appreciate author efforts. However, this approach has no technical reasons to be more sensitive compared to multiplex PCR, the authors explanations provided are not persuading for me.

The main explanation provided is that: "PCR-based methods is their inability to accurately annotate J regions". This obviously has no practical justification, since multiplex amplification from the C region does not affect the J region in any way. Appropriate parameters in data analysis are the key here.

Reviewer #4 (Remarks to the Author):

I recommend publication of this manuscript

Point by Point:

Reviewer #1 (Remarks to the Author):

I think the revised manuscript has substantially addressed most of my concerns and significantly improved. The authors did a comparison of SPTCR-seq with other spatial-TCR methods, and reconstructed more TCR clonotypes per sample by their method, though with higher cost and longer hands-on time. A head-to-head comparison of various methods on same samples is definitely welcome, although I have reservation on the design and conclusions made in this manuscript. The authors explained that the differences of performance mostly come from the long read-length of SPTCR-seq and shorter read-length of multiplex-PCR based methods, however, the PCR based method can also be adapted to long read length by nanopore sequencing. Therefore, it is an unfair comparison and should be noted in the manuscript.

We greatly appreciate the reviewer's thoughtful feedback and the opportunity to clarify the aspects of our study related to the comparison of SPTCR-seq with other spatial-TCR methods.

We concur with the reviewer's assessment that the use of long read sequencing in our SPTCR-seq method could introduce a confounding element in our comparative analysis. The goal of this experimental design was to maintain protocol consistency across all methods under evaluation, as our primary objective was to perform an apples-to-apples comparison by keeping the protocol uniform, thus minimizing variation introduced by different sequencing methods.

However, we understand the reviewer's concern that this approach may not fully reflect the potential of multiplex-PCR based methods when adapted to long read length by nanopore sequencing. Given the constraints of our study, a comprehensive comparison involving all different sequencing methods across all protocols was beyond the scope of our investigation. We intended our work to serve as an initial comparative evaluation rather than a definitive benchmarking study. Yet, we acknowledge that the impact of read length variations between methods could have been more extensively discussed, and the potential for PCR-based methods to also employ long read sequencing could have been acknowledged. To address the reviewer's concern, we will make modifications to our manuscript to more explicitly note that the comparison we made is largely dependent on the specific sequencing protocols used. This includes the impact of using long read sequencing exclusively with our SPTCR-seq method, while short read sequencing was used in other methods. We will also highlight the potential for adapting PCR-based methods to long read length sequencing, noting that this adaptation could possibly lead to a different outcome in the comparison.

"... Further, we acknowledge that our comparative analysis was primarily influenced by the specific sequencing protocols utilized, particularly the exclusive use of long read sequencing for our SPTCR-seq method. Future studies should explore the potential of adapting multiplex-PCR methods to long read sequencing, as this could significantly shift the performance landscape of various spatial-TCR methods."

The authors added substantial analysis of adopting spatially-resolved TCR data and other multi-omics data, such as transcriptome and metabolome data, to link the T cell expansion and diversity with patterns of cellular interactions, as well as T cell dysfunction or cytotoxicity. I think these analyses added more value in the revised manuscript.

We thank the reviewer for his comment.

Some details are listed below.

1. L283-286, there are no supporting data/figures for these results.

We added a boxplot indicating the T cell specific expression of exhausted and cytotoxic gene expression across clonal or diverse TCR distribution.

2. L350-351, citations should be added here.

We added the missing citation

3. L373-374, description of “additional layer of metabolomics” should be added, or a proper citation with the detailed description can be added.

We changed the sentence:

“... To this end, we added an additional molecular layer of spatially resolved metabolomics (MALDI, n=5) as recently described⁴, to functionally characterize identified T cell clones. ...”

4. L415-417, these sentences should be rephrased, as noted by the argument I mentioned above, also, PCR cycle reduction does not lead to higher sensitivity?

We removed the statement: “..., Additionally, SPTCR-seq effectively minimizes PCR amplification cycles, identified as a major source of sequencing errors in our analysis. In contrast to the alternative

methods, Hudson's (70 [35+35] PCR cycles) and Liu's (30 [18+12] PCR cycles), SPTCR-seq necessitates a total of only 23 (5+18) PCR cycles. This reduction in PCR cycles, coupled with consensus-based error correction, enhances sequence accuracy and contributes to more sensitive and specific TCR detection. “

Reviewer #3 (Remarks to the Author):

I appreciate author efforts. However, this approach has no technical reasons to be more sensitive compared to multiplex PCR, the authors explanations provided are not persuading for me. The main explanation provided is that: "PCR-based methods is their inability to accurately annotate J regions". This obviously has no practical justification, since multiplex amplification from the C region does not affect the J region in any way. Appropriate parameters in data analysis are the key here.

Our comparison of all methods revealed that “...inability to accurately annotate J regions...” was the main problem of the PCR-based methods.

Reviewer #4 (Remarks to the Author):

I recommend publication of this manuscript